# Arctic amplification is caused by sea-ice loss under increasing $CO_2$

Aiguo Dai [1], Dehai Luo[2], Mirong Song[3] & Jiping Liu[1]

Warming in the Arctic has been much faster than the rest of the world in both observations and model simulations, a phenomenon known as the Arctic amplification (AA) whose cause is still under debate. By analyzing data and model simulations, here we show that large AA occurs only from October to April and only over areas with significant sea-ice loss. AA largely disappears when Arctic sea ice is fixed or melts away. Periods with larger AA are associated with larger sea-ice loss, and models with bigger sea-ice loss produce larger AA. Increased outgoing longwave radiation and heat fluxes from the newly opened waters cause AA, whereas all other processes can only indirectly contribute to AA by melting sea-ice. We conclude that sea-ice loss is necessary for the existence of large AA and that models need to simulate Arctic sea ice realistically in order to correctly simulate Arctic warming under increasing $CO_2$.

[1] Department of Atmospheric & Environmental Sciences, University at Albany, SUNY, Albany, NY 12222, USA. [2] CAS RCE-TEA, Institute of Atmospheric Physics, Chinese Academy of Sciences, Beijing 100029, China. [3] LASG, Institute of Atmospheric Physics, Chinese Academy of Sciences, Beijing 100029, China. Correspondence and requests for materials should be addressed to A.D. (email: adai@albany.edu) or to D.L. (email: ldh@mail.iap.ac.cn)

Enhanced warming in the Arctic (north of 67°N) is seen in recent observations[1–5] and model simulations[6–9] with increasing greenhouse gases (GHGs), a phenomenon referred to as the Arctic amplification (AA)[1,10] that reduces meridional temperature gradients and thus may affect mid-latitude weather and climate[5,11–17]. Many mechanisms have been proposed to explain the AA, including a central role by sea-ice loss[2,3,18,19], reduced outgoing longwave (LW) radiation due to a stable polar temperature profile[20,21], increased downward LW heating due to increased water vapor and clouds[19,22,23], increased poleward energy transport[23,24], and other processes[25–33]. However, their relative importance is still under debate. In particular, why the largest AA occurs in the cold season (when the ice–albedo effect is small) and over areas with large sea-ice loss[3] has not been well explained by these mechanisms.

Seasonal sea-ice melting from May to September opens a large portion of the Arctic Ocean, allowing it to absorb sunlight during the warm season. Most of this energy is released to the atmosphere through longwave (LW) radiation, and latent and sensible heat fluxes during the cold season from October to April when the Arctic Ocean becomes a heat source to the atmosphere[10] (Supplementary Figure 1). Under greenhouse gas (GHG)-induced global warming, Arctic sea ice is expected to decrease greatly[7,8,34], which increases the absorption of sunlight by the Arctic Ocean during the warm season and its subsequent release as more Arctic Ocean becomes ice-free, thereby amplifying Arctic warming in the cold season. This process involves the seasonal storage and release of the absorbed solar radiation, thus it differs from the ice–albedo feedback over land and is not the same as just altering the surface albedo in climate models[35,36], as noticed previously[37]. While this process has been recognized[10,38] and examined using recent reanalysis data[28], its impact on the AA under increasing GHGs is not fully understood and needs further investigation. Furthermore, it is unclear whether and how the other processes mentioned above are related to sea-ice loss in producing the AA. Is sea-ice loss necessary for large AA to occur? Can the reduced LW cooling[20,21] or the increased LW heating[19,22] still produce large AA under GHG-induced global warming without significant sea-ice loss? Can the seasonal and spatial patterns[3] of the AA be explained by any of the proposed mechanisms?

We address these questions here by analyzing the historical (1979–2016) and future (up to year 2300) changes in Arctic sea-ice cover (SIC), surface air temperature (Tas), and energy fluxes in ERA-Interim reanalysis data and CMIP5 model simulations (see Methods). We also perform and analyze two climate change simulations with 1%-per-year increase in atmospheric $CO_2$ with and without fixed SIC in calculating surface water and energy fluxes using a fully coupled climate model (namely, the CESM1, see Methods). These models realistically simulate the mean annual cycle of SIC, Tas, and surface energy fluxes (Supplementary Figure 1), as well as the SIC distributions (Supplementary Figures 2–3). In the CMIP5 simulations, Arctic sea-ice loss rate peaks around 2070; thereafter, both Arctic SIC and sea-ice loss diminish while global warming continues, albeit at a slower pace after the late 22nd century due to reduced GHG forcing[39] (Supplementary Figure 4). This allows us to examine how the magnitude of the AA varies with changing SIC and sea-ice loss, and whether the existence of sea ice is necessary for large AA to occur. In the CESM1 simulations, a constant $CO_2$ forcing with a 1%-per-year increase is applied for 235 years during which atmospheric $CO_2$ level is doubled three times at year 70, 140, and 210. These simulations allow us to examine how the AA varies over time as the SIC changes under a constant external forcing, and the difference between the 1% $CO_2$ run with fully interactive sea ice (denoted as 1% $CO_2$ run) and the 1% $CO_2$ run with fixed SIC in calculating the surface fluxes (denoted as FixedIce run)

would allow us to further quantifying the impact of sea-ice loss on AA through its impact on surface fluxes.

We found that large AA occurs only from October to April and only over areas with significant sea-ice loss in both observations and model simulations. AA largely disappears when Arctic sea ice melts away or is held fixed for calculating surface fluxes. Periods with large AA are associated with large sea-ice loss in model simulations, and models with bigger sea-ice loss produce larger AA. Increased LW radiation and latent and sensible heat fluxes from the newly exposed Arctic waters enhance surface and low-tropospheric warming and cause AA, whereas water vapor feedback, increased downward LW radiation, and other processes can only modulate the AA induced by sea-ice loss or indirectly contribute to AA by melting sea ice. Our results highlight the essential role of sea-ice loss in producing AA under GHG-induced global warming.

## Results
**Historical changes**. From 1979–2016, Arctic SIC has decreased considerably in all months, especially during June–November, which leads to increased absorption of solar radiation by the Arctic from April to September (Fig. 1a). However, the largest AA (defined here as the ratio of Arctic vs. global-mean Tas change) occurs from October to April, while the AA is small during the warm season, especially in July–August (Fig. 1a). The extra absorbed solar radiation during the warm season occurs over and is stored in the newly opened Arctic waters with minimum enhancement of surface temperatures as reflected by the small changes in surface upward LW radiation (LW_up) and sensible (SH) and latent (LH) heat fluxes during the warm season (Fig. 1a). From October to April, LW_up and SH + LH fluxes have increased substantially, which indicates a warm ocean surface and extra heating to the air. This leads to enhanced atmospheric warming near the surface (Fig. 1a) and in the lower troposphere[3]. The November-minus-July difference in the LW_up ($\sim 4$ W/m$^2$/decade) and SH + LH ($\sim 1.7$ W/m$^2$/decade) fluxes in Fig. 1a may be attributed to the amplified ocean surface warming (due to a combination of the extra solar absorption during the warm season and the opening of new water surfaces during the cold season). Clearly, the LW_up forcing of the air is more than twice the SH + LW change, which is also the case in the model simulations discussed below. This is expected because of the large (10–30 °C) temperature difference between typical water and ice surfaces in the winter Arctic[40] (Supplementary Figure 3). Such a scenario is also supported by the close spatial collocation of the largest surface warming and turbulent flux (and LW_up) increases with the largest sea-ice loss (Fig. 2a). These historical changes, including the seasonality and spatial patterns, are largely reproduced by the CMIP5 models (Figs. 1b, 2b). Because the CMIP5 multi-model ensemble mean represents mostly externally forced changes[41,42], this result suggests that these historical Arctic changes are mostly forced by $CO_2$ and other external forcing. It also implies that the CMIP5 models may be capable of simulating the Arctic responses to $CO_2$ and other greenhouse gas changes under future forcing scenarios.

**CMIP5 model-projected changes**. CMIP5 model-simulated Arctic warming and sea-ice loss vary greatly with month in the 21st and 22nd centuries, but not in the 23rd century when most of the sea ice is melted away (Fig. 3). During the 21st century, large sea-ice loss (>20% of the Arctic area) occurs from June to January, but large AA exists only from October to April, peaking in November–December (Fig. 3a). Reduced SIC allows the Arctic Ocean to absorb more sunlight from April to August (Fig. 3a), but this extra energy is stored in the upper Arctic Ocean without

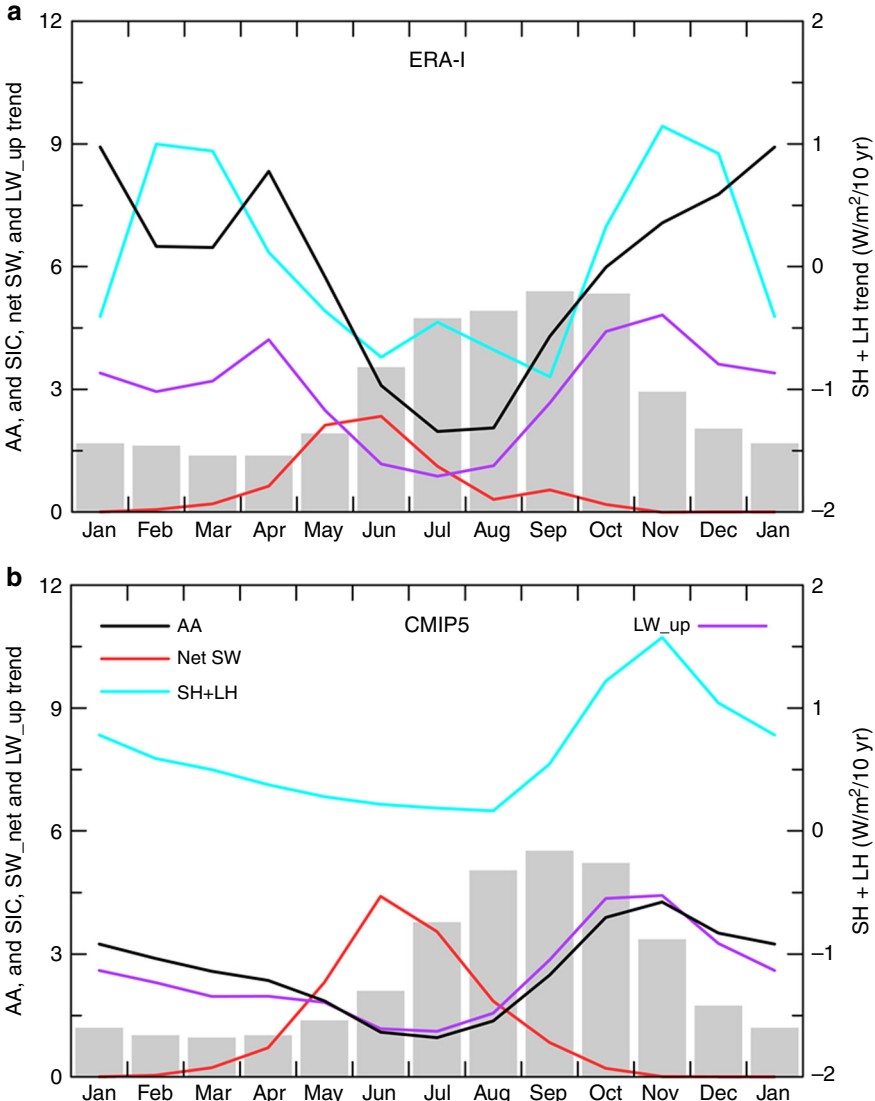

**Fig. 1** Seasonality of the historical (1979–2016) trends in Arctic (67°–90°N) sea-ice cover (SIC), Arctic amplification (AA), and Arctic energy fluxes. **a** from ERA-Interim reanalysis data and **b** from the ensemble mean of historical (for 1979–2005) and RCP8.5 (for 2006–2016) simulations averaged over 38 CMIP5 models. The SIC trend (gray bars) is in $10^5$ km²/decade; the AA (black line) is defined as the ratio of the surface air temperature trends between the Arctic and the globe; the surface net shortwave (red, positive downward), sensible plus latent heat (blue, positive upward) and upward longwave (magenta) flux trends are in W/m²/decade

increasing the surface temperatures substantially due to the large heat capacity of the ocean mixed layer. This results in small changes in LW_up and SH + LH fluxes and negligible AA for the summer months, consistent with the recent changes[3,28] (Fig. 1a). This result suggests that when the Arctic Ocean is a heat sink from May to August (Supplementary Figure 5a), all the surface and atmospheric changes (including increased LW heating from increased water vapor and clouds, Supplementary Figure 6) cannot produce AA during those months. However, from October to March when the Arctic Ocean becomes a heat source to the atmosphere (Supplementary Figure 5a), the extra energy stored in the ocean is released through surface upward LW radiation, LH and SH fluxes to heat the lower troposphere, thereby enhancing the Arctic warming during these months (Fig. 4a), as LW radiation and SH directly warm the lower troposphere while LH increases water vapor and thus its greenhouse warming effect on the surface. This key role by surface LW, LH, and SH fluxes is consistent with that seen during the recent decades[28] (Fig. 1). Clearly, whether an area is covered by sea ice during the cold

season makes a huge difference for these surface fluxes, and this is why sea-ice loss can greatly enhance the warming induced by $CO_2$ and water vapor increases.

By the end of the 21st century, Arctic sea ice is largely gone from July to October (Supplementary Figure 5b). As a result, the largest sea-ice loss in the 22nd century occurs from December to June, and the largest increases in the absorbed SW radiation are from April to June (Fig. 3b). Again, this extra energy is stored in the Arctic Ocean with negligible amplification of surface warming during these months (Fig. 3b). As the Arctic sea ice continues to decline during the cold season (Fig. 3b), more ocean surfaces are ice-free and open to the atmosphere. Because the newly exposed ocean water is much warmer than the cold sea-ice surface existed previously[40], this allows the ocean to release more LH and SH fluxes and LW radiation (Fig. 3b) to warm the lower troposphere from November to March, when the ocean is still a seasonal heat source to the atmosphere (Supplementary Figure 5b). The largest release of the extra LW, LH, and SH is delayed to January–February by the end of 22nd century from November

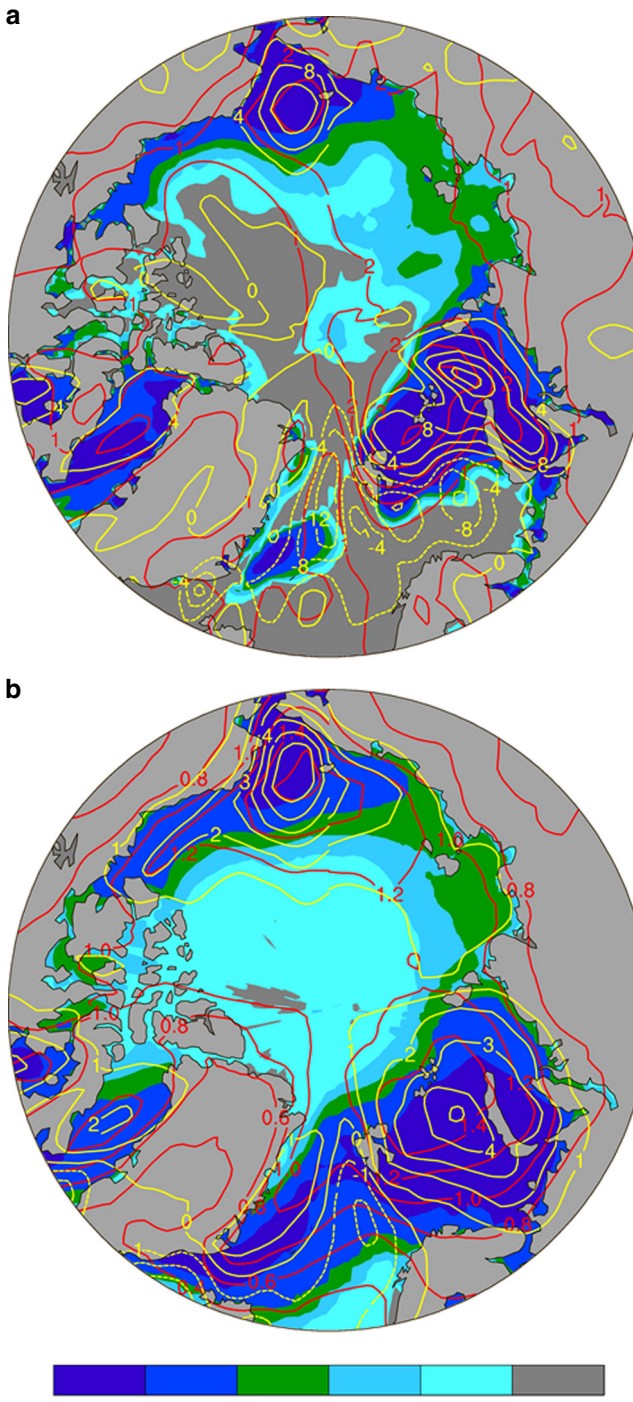

to December in the 21st century as the maximum sea-ice loss moves to later months (Fig. 3a, b). This results in elevated Arctic warming and thus large AA from November to March, whereas the AA is small for the other months (Fig. 3b).

During the 23rd century, sea ice forms only over a small area (<20%) of the Arctic Ocean even during the cold season (Supplementary Figure 5c, d). Thus, there is no significant sea-ice loss for most of the months except for January–May when small (3–10%) losses still occur (Fig. 3c). Because of this, absorbed solar radiation changes little, as do the surface LH and SH fluxes; while the upward LW increases uniformly throughout the year due to the overall surface warming (Fig. 3c). Without the extra heating from the ocean (on top of the mean seasonal cycle, Supplementary Figure 5c, d), the Arctic warming during the 23rd century shows little seasonal variation, and it is only about 40–50% higher than the global warming rate for all the months (Fig. 3c), which is comparable to the AA during the summer in previous centuries (Fig. 3a, b).

We further calculated the local change between two moving 20-yr periods for Arctic annual SIC, Arctic and global-mean annual Tas, and the Arctic-minus-global warming difference (Fig. 3d). Clearly, the elevated warming over the Arctic is strongly correlated with sea-ice loss (r = 0.95) for all the 40-yr periods within 1900–2300. As the Arctic sea-ice loss increases from the 1950s to the 2060 s, the Arctic-minus-global warming difference also increases; thereafter, the warming difference decreases as the sea-ice loss slows down due to reduced SIC (Fig. 3d). After year 2200, there is little sea-ice left to melt, and the warming difference is small between the Arctic and the globe. In contrast, from 1980–2020 when the global warming rate is similar to that of 2200–2280, the Arctic warms much faster than the global-mean as the sea-ice loss is much larger during 1980–2020 (Fig. 3d).

We also found a strong correlation (r = 0.84) between the Arctic sea-ice loss and Arctic-minus-global warming difference from 1979 to 1999 and from 2070 to 2099 among the 38 CMIP5 models we analyzed (Fig. 4); that is, models with a larger sea-ice loss tend to produce larger AA in the 21st century under the RCP8.5 scenario. Since the current SIC determines how much sea ice is available for future melting, this result suggests that the magnitude of a model-simulated Arctic warming and AA will depend on its mean bias in SIC (and thus also Tas) for the current climate, consistent with previous studies[6,33].

Strong spatial pattern correlations were also found between Arctic sea-ice loss and surface warming and changes in surface energy fluxes (Fig. 5), consistent with recent observations[3,28] (Fig. 2a). For example, large surface warming and large increases in LH and SH (and LW_up) fluxes in December are collocated with large sea-ice loss in both the 21st and 22nd centuries (Fig. 5), while the enhanced warming largely disappears in the 23rd century when the sea ice is gone (Supplementary Figure 7). Without a central role of sea-ice loss, it would be difficult to think of any mechanism for reduced LW cooling due to a stable temperature profile[20,21], increased LW heating from increased water vapor and clouds[19,22,23,43], increased poleward heat transport[23,24,43], or other processes to generate such a spatial pattern of surface warming that resembles that of sea-ice loss and then their effects disappear in the 23rd century when there is little sea-ice loss. While enhanced local warming can increase sea-ice loss and thus cause a negative correlation between the SIC and Tas change, the warming pattern itself cannot be easily explained by the LW- and heat transport-related mechanisms without a key role of the sea-ice loss. This is because the large-scale downward LW forcing (from increased $CO_2$, water vapor, or clouds) should not be correlated spatially with SIC loss unless there exists a major role by the sea-ice loss to alter the surface warming and water vapor (and thus LW) change patterns. In other words, if

**Fig. 2** Spatial distributions of the linear trends during 1979–2016. For November–December mean surface air temperature (red contours, K/decade), sea-ice concentration (SIC, color shading, %/decade), and surface turbulent (sensible + latent) heat fluxes (yellow contours, W/m²/decade, positive upward) based on **a** the ERA-Interim reanalysis data and **b** the ensemble mean of historical (for 1979–2005) and RCP8.5 (for 2006–2016) simulations averaged over 38 CMIP5 models. The SIC trends are similar to those based on NOAA satellite data from https://sidads.colorado.edu/DATASETS/NOAA/G02202_V3. Spatial pattern correlations: Trend pattern correlations: r (SIC,Tas) = −0.61, r (SIC,LH + SH) = −0.68, r (Tas, LH + SH) = 0.56 in **a**; and r (SIC,Tas) = −0.40, r (SIC,LH + SH) = −0.70, r (Tas,LH + SH) = 0.65 in **b**. The upward longwave radiation trend (not shown) is highly correlated with the air temperature trend (r ≥ 0.96). These correlations have a p-value well below 0.01

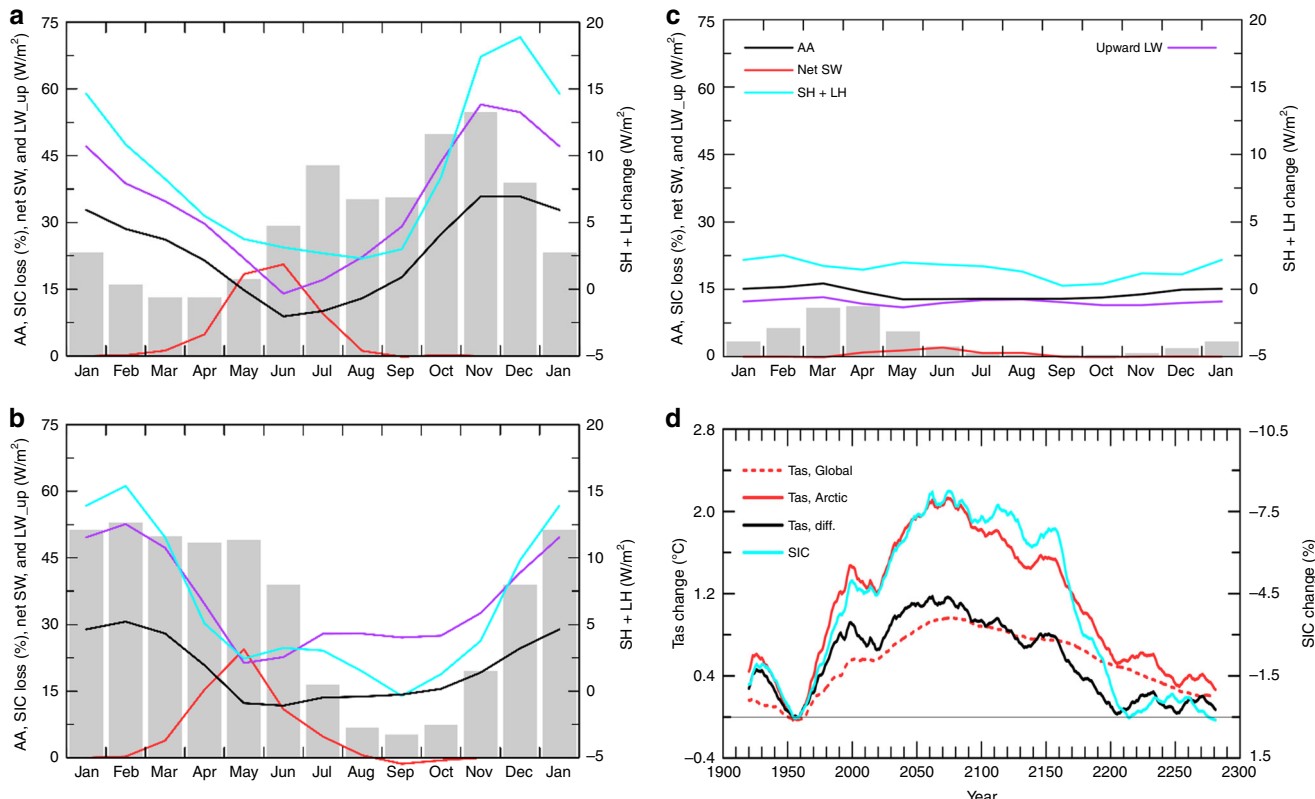

**Fig. 3** Centennial changes as a function of month from CMIP5 models. For Arctic (67°−90°N) sea-ice concertraion (SIC, in % of Arctic area, shading, multiplied by −1), Arctic-to-global ratio of the Tas change (AA, black line, multiplied by 10 in order to use the left y-axis), and Arctic surface energy fluxes (in W m⁻²). **a** 2070–2099 minus 1970–1999, **b** 2170–2199 minus 2070–2099, and **c** 2270–2299 minus 2170–2199 under the historical and RCP85 scenarios from the ensemble mean of nine model runs from the nine CMIP5 models. Net SW = net shortwave radiation (positive downard), upward LW = upward longwave radiation, SH = sensible heat, LH = latent heat. **d** Time-dependent warming and sea-ice loss from CMIP5 models. Time series of the difference between the 20 year periods separated by the plotted year in annual Arctic (67°−90°N, red solid) and global-mean (red dashed) surface air temperature (Tas), annual Arctic SIC (blue), and the difference between the red solid and red dashed lines (black) based on the ensemble mean of nine simulations from nine CMIP5 models. The correlation coefficient between the lines are: r (SIC, Tas_Arctic) = −0.97, r (sic,Tas_global) = −0.90, r (sic, Tas_diff) = −0.95, and r (sic, AA) = −0.80, where AA = the ratio of the Arctic to global Tas change (data before 2000 were not used for AA). These correlations have a *p*-value well below 0.01

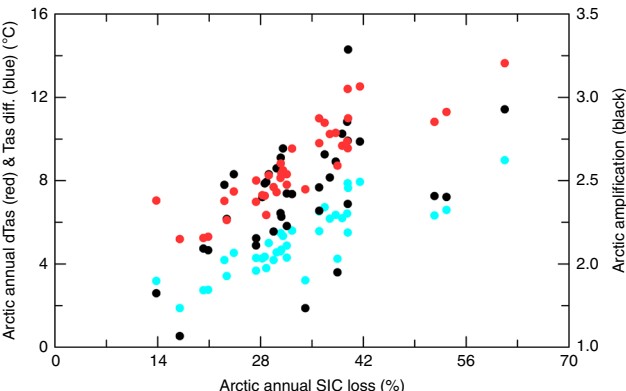

**Fig. 4** Dependence of Arctic warming and amplification on sea-ice loss among 38 CMIP5 models. Scatter plot of 2070–2099 minus 1970–1999 difference under the RCP85 scenario between annual Arctic SIC loss and Arctic surface warming (red), Arctic-minus-global warming difference (blue), or the Arctic-to-global warming ratio (i.e., the Arctic amplification, or AA, black). Each dot is for one CMIP5 model. Correlation coefficients: r(SIC, dTas_Arctic) = 0.87 (*p* = 0.00), r(SIC, Tas_diff) = 0.84 (=0.00), and r (SIC, AA) = 0.54 (*p* = 0.01)

SIC does not play a major role, then the warming pattern should be fairly uniform (as $CO_2$ and water vapor in the Arctic would be well mixed zonally by the large-scale circulation if local processes played a minimal role), or spatially correlated with cloudiness changes rather than with SIC loss. Thus, while increased downward LW radiation associated with increased $CO_2$ and water vapor may play a large role for overall Arctic[23,43] and global[44] warming, it cannot produce the enhanced warming collocated with sea-ice loss and existed primarily only during the cold season that leads to large AA only in the cold season. However, a general Arctic warming, either due to increased $CO_2$, water vapor, or clouds, can indirectly contribute to AA through melting of sea ice.

On the other hand, the increased LW_up, SH, and LH heating by the newly opened Arctic waters (Fig. 5) should increase lower tropospheric temperature and water vapor, and possibly cloud cover as well (Supplementary Figure 6) over and around the areas with sea-ice loss. These atmospheric changes, triggered by the extra surface heating induced by sea-ice loss and enhanced by the local positive water vapor feedback, result in increased downward LW radiation (Supplementary Figure 6) as noticed previously[19,22,23], which in turn helps maintain and enhance an elevated surface temperature over the newly opened waters. In turn, the enhanced warming should accelerate the sea-ice loss, leading to a positive feedback loop. The stable Arctic

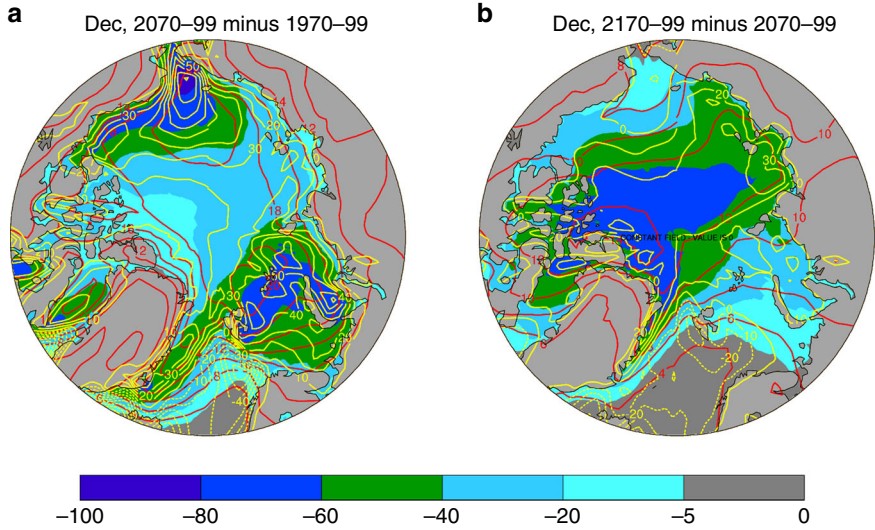

**Fig. 5** Centennial changes from CMIP5 models. **a** from 1970–1999 to 2070–2099 and **b** from 2070–2099 to 2170–2199. Shown are changes in December sea-ice concentration (SIC, %, color shading), air temperature (Tas, °C, red contours, interval = 2), and latent and sensible heat fluxes (LH + SH, W m$^{-2}$, yellow contours, interval = 10). Dashed contours are for negative values. Based on the ensemble mean of nine simulations from nine CMIP5 models under the historical and RCP85 scenarios. The spatial pattern correlations are: r (SIC, Tas) = −0.79, r (SIC, LH + SH) = −0.68, r (Tas, LH + SH) = 0.62 in **a**, and r(SIC, Tas) = −0.85, r (SIC,LH + SH) = −0.65, r (Tas, LH + SH) = 0.60 in **b**. Surface net energy flux change (not shown) is similarly correlated with the SIC and Tas changes, while the upward LW flux change (not shown) is highly correlated (r≈0.90) with the Tas change. These correlations have a *p*-value well below 0.01

atmosphere[45] may also allow the extra surface heating to generate large warming in the lower troposphere by reducing LW cooling to space[20,21]. However, the fact that the AA becomes much smaller in the 23rd century (October–April mean Arctic to global warming ratio = 1.47) than that in the 21st (2.97) and 22nd (2.40) century (Fig. 3) suggests that the LW, latent and sensible heating due to sea-ice loss is essential for large AA to occur, while all the other processes can only modulate the sea-ice loss-induced amplification or indirectly contribute to AA by causing sea-ice melting, but they cannot cause large AA without sea-ice loss.

**Results from CESM1 simulations**. One concern with the extended CMIP5 simulations is that the mean climate in the 23rd century is much warmer than and likely very different from today's climate and this may become an important factor for AA, although the seasonal, spatial, temporal, and inter-model dependence of the AA on sea-ice loss is seen during all time periods besides the 23rd century in the CMIP5 simulations. Another concern is that the external forcing and global warming rate change during the CMIP5 simulations, which may affect the AA. To address these concerns and to further isolate and quantify the effect from sea-ice loss, we performed two multi-century simulations with a constant forcing of 1%-per-year $CO_2$ increases using NCAR CESM1, a state-of-the art fully coupled climate model. The first simulation is the standard 1%/year $CO_2$ run with fully active sea ice (1% $CO_2$ run), and the second run is the same except it uses fixed Arctic sea-ice cover in *calculating* all the surface fluxes only (FixedIce run; see Methods). Thus, the 1% $CO_2$-minus-FixedIce difference represents the effect of Arctic sea-ice loss through its impact on surface fluxes, and a consistent dependence of the AA on sea-ice loss during these multi-century simulations would indicate a weak dependence of the AA on the mean climate state, as these simulations cover a wide range of atmospheric $CO_2$ from 284.7 to 2950 ppm (and thus of the mean climate). Without the sea-ice loss-induced changes in surface fluxes, Arctic warming is greatly reduced while global warming weakens only slightly under the 1%-per-year $CO_2$ increase, and this results in negligible AA for annual-mean Tas during the

entire simulation (Fig. 6). In fact, the Arctic-global Tas difference becomes negative after about year 160 in the FixedIce run (Fig. 6b). This strongly suggests that without the sea-ice loss-induced surface flux changes, no physical mechanisms can cause AA for annual-mean Tas.

Similar to the CMIP5 results (Fig. 3), the large sea-ice loss in the CESM1 1% $CO_2$ run also starts mainly during the summer–early winter season (Fig. 7a, b), and then extends to the winter–spring season (Fig. 7c) as the warming intensifies under increasing $CO_2$. Again, the sea-ice loss and other changes do not cause large AA during the summer months throughout the simulation (Figs. 6a, 7), whereas large AA is seen during the cold season when the releases of LW, SH, and LH fluxes into the Arctic air increase (Fig. 7a–c). The changes over a moving 40-year period (Fig. 7d) also show strong negative correlation (r = −0.87) between the local SIC change and the Arctic-global Tas difference, again suggesting a vital role of the sea-ice loss for AA.

When a constant sea-ice cover is used in computing the surface fluxes, Arctic sea-ice loss is greatly reduced (Fig. 8a–c) as a result of the reduced Arctic warming (Fig. 6b). The reduced sea-ice loss contributes to small changes in surface LW_up, SH, and LH fluxes and small AA even during the cold season throughout the simulation (Figs. 6b, 8). The Arctic warming during the warm season is actually smaller than the global mean (Figs. 8c, 9e), likely because the $CO_2$-induced warming is lower over ocean water surfaces than over continents, and the Arctic Ocean is essentially ice-free during the summer months by the 3rd $CO_2$ doubling. As a result, the annual-mean Tas shows very little AA even by the time of the 3rd $CO_2$ doubling (Fig. 6b). With sea-ice loss in the 1% $CO_2$ run, warming in the lower troposphere is larger over the Arctic than the midlatitudes for the cold season and annual mean (Fig. 9a, c), which would reduce meridional temperature gradients in the lower troposphere over the northern latitudes and therefore could potentially weaken midlatitudes westerlies and jet stream[15–17]. Such an effect is absent, however, in the FixedIce run (Fig. 9d–f). Thus, sea-ice loss is necessary for GHG-induced warming to alter the meridional temperature gradient and thus affect the weather and climate at northern midlatitudes.

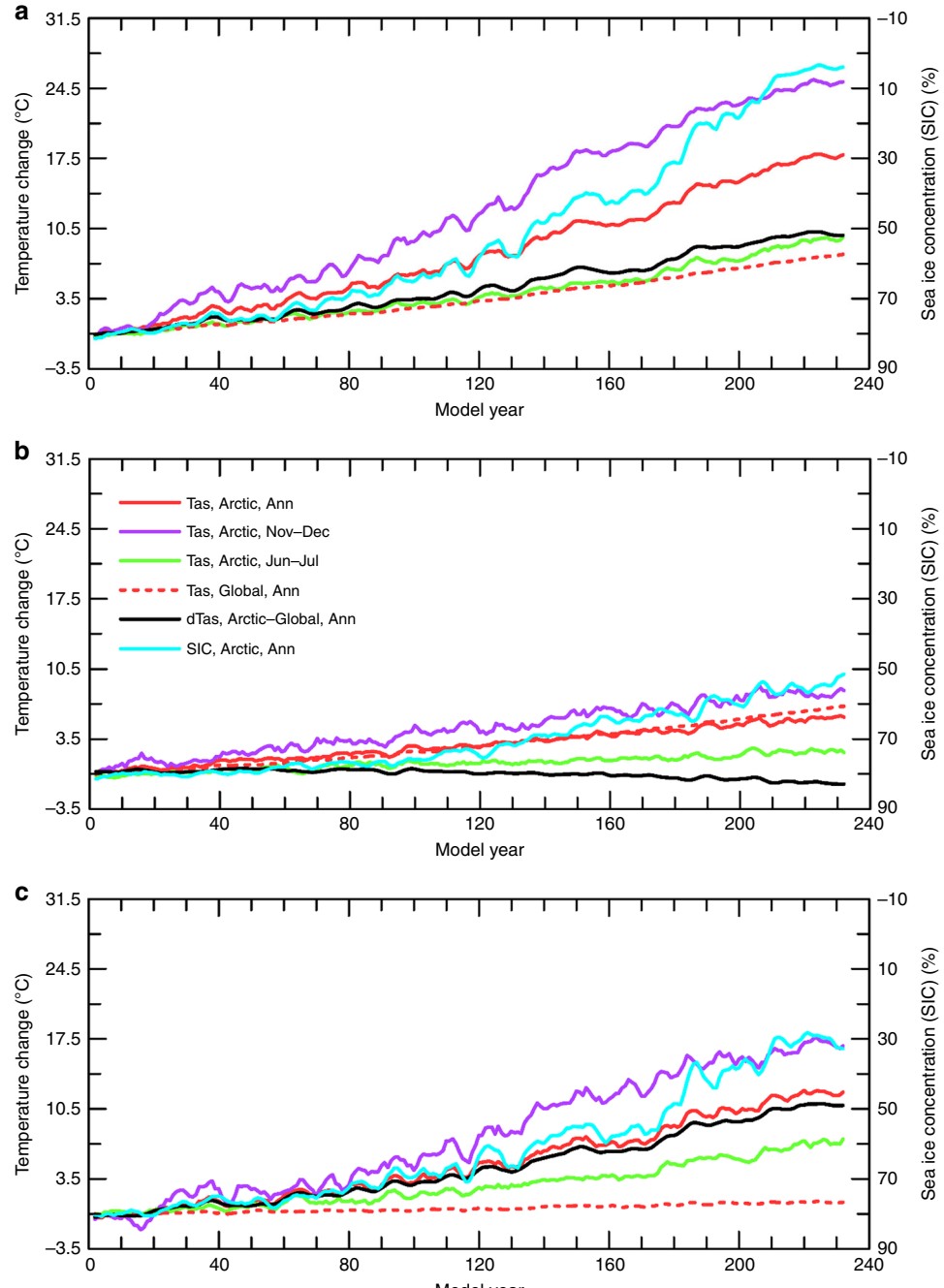

**Fig. 6** Time series of CESM1-simulated changes in surface air temperature (Tas) and sea-ice over the Arctic (67°-90°N) and globe. **a** Standard 1% CO$_2$ run, **b** FixedIce run, and **c** their difference (panel **a** minus panel **b**). Shown are the annual (solid red), November–December (magenta), and June–July (green) mean Tas, and annual Arctic sea-ice concentration (blue, right y-axis, increase downward), together with the Arctic-minus-global annual Tas difference (black). The change is relative to the control-run climatology and five-year averaging is applied. Note that global Tas changes for November–December and June–July (not shown) are very similar to the annual change

The relationship between local changes (Fig. 8d) in sea-ice loss and the Arctic-global Tas difference becomes weaker as their physical relationship is altered by the use of a constant SIC in calculating the surface fluxes in the FixedIce run. This weakened coupling between surface warming and sea-ice loss is also seen in the spatial correspondence (Fig. 10b), in contrast to the fully coupled run (Fig. 10a) and the CMIP5 simulations (Fig. 5), in which large releases of SH and LH fluxes are collocated with the large sea-ice loss and enhanced surface warming. Without the sea-ice loss-induced flux changes, surface warming is quite uniform spatially over the Arctic and comparable with that over

the lower latitudes (Fig. 10b). This again implies an essential role of sea-ice loss through its impact on surface fluxes (mainly LW_up, SH, and LH) in producing a large AA.

## Discussion
The seasonal, temporal, inter-model, and spatial dependence of the AA on sea-ice loss presented above, coupled with the differences between the CESM1 1% CO$_2$ and FixedIce runs, strongly suggests that the existence of sea ice and significant sea-ice loss is necessary for large AA to occur under GHG-induced warming,

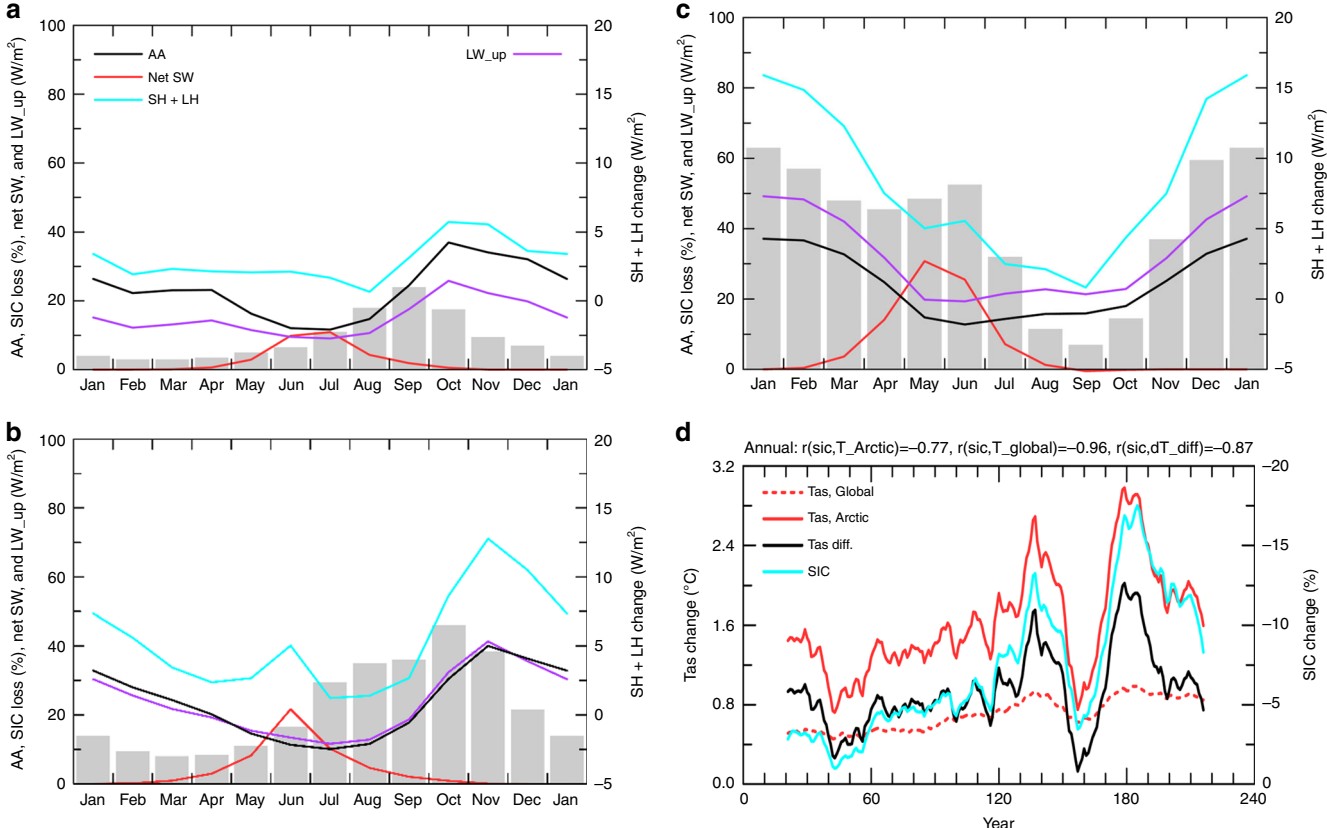

**Fig. 7** Centennial changes as a function of month from CESM1 standard simulation with a 1%-per-year $CO_2$ increase. Same as Fig. 3, except for the change during the **a** first (year 61–80 mean minus control climatology), **b** second (year 131–150 mean minus year 61–80 mean), and **c** third (year 201–220 mean minus year 131–150 mean) $CO_2$ doubling from the standard simulation with a 1% $CO_2$ increase per year using the CESM1. **d** Time-dependent warming and sea-ice loss from CESM1 simulation with a 1%-per-year $CO_2$ increase. Same as Fig. 3d except for the CESM1 1% $CO_2$ run

and that AA occurs primarily in the cold season due to the extra LW radiation and sensible and latent heat release from the newly opened waters, which are 10–30 °C warmer than sea-ice surfaces in Arctic winter[40] (Supplementary Figure 3). The results also suggest that water vapor and other feedbacks can only help maintain and enhance the AA triggered by sea-ice loss or indirectly contribute to AA by causing sea-ice melting, but they cannot produce large AA without sea-ice loss. Large AA cannot occur during the warm season because any extra heating will be absorbed by and stored in the upper Arctic Ocean, rather than being used to raise air temperature. We emphasize that enhanced downward LW radiation[23] from increased $CO_2$, water vapor, and other greenhouse gases will lead to Arctic and global warming, which drives the decline of Arctic sea ice on interannual[43], decadal[23], and longer[44] time scales. Poleward energy transport is essential for Arctic energy balance, and increased such transport in a warmer climate[23,24] would help the Arctic stay in a warmer state, contributing to sea-ice melting. The sea-ice loss in turn causes AA under a warming climate. From this perspective, any external forcing (e.g., $CO_2$ increases) or internal processes (e.g., water vapor feedback[19,22], poleward heat transport[24], a stable lower troposphere[20]) that can cause or enhance Arctic warming may contribute indirectly to AA through their impact on sea ice. However, without the LW, SH, LH, and other surface flux changes associated with the sea-ice loss (e.g., after the sea ice melts away as in past or future warm climates, or in a model with little sea-ice loss), Arctic warming rate under increasing GHGs would be similar to the global warming rate, leading to small AA.

Different from the response to GHG forcing analyzed here, Arctic warming in observations (especially over short periods)[23,43]

and individual model simulations may also include decadal–multidecadal changes induced by internal climate variability[30,32]. It is unclear whether sea-ice loss also plays such a central role for the AA induced by internal variability, although there is no reason to prevent the mechanism presented above from working under this case. Modeling studies[35,36] with fixed surface albedo still produced noticeable AA, in which LW_up, SH, and LH fluxes could still change as SIC declines. This further points to the key role of increased LW_up, SH, and LH fluxes associated with sea-ice loss, besides the albedo effect which occurs mainly in the warm season and thus cannot directly affect the AA, which occurs primarily in the cold season.

We conclude that sea-ice loss is necessary for large AA to occur because it mainly results from the increased upward LW radiation and SH and LH fluxes during the cold season over the newly opened waters, while water vapor and other feedbacks can only enhance and maintain the surface warming triggered by sea-ice loss or indirectly contribute to AA by causing sea-ice melting. While increased downward LW radiation can contribute to Arctic warming, it cannot produce an *amplified* warming over the Arctic compared with the rest of the world if there is no sea-ice loss. On the other hand, how important of the additional absorption of solar radiation (due to ice–albedo effect) during the warm season (red line in Fig. 3a, b) is for the enhanced release of the LW, SH, and LH fluxes during the cold season and thus to the AA requires further investigation.

## Methods
**CMIP5 simulations**. The monthly model data were obtained from 38 historical (1900 to 2005) and RCP8.5 (2006 to 2100) simulations from 38 CMIP5 models[46]

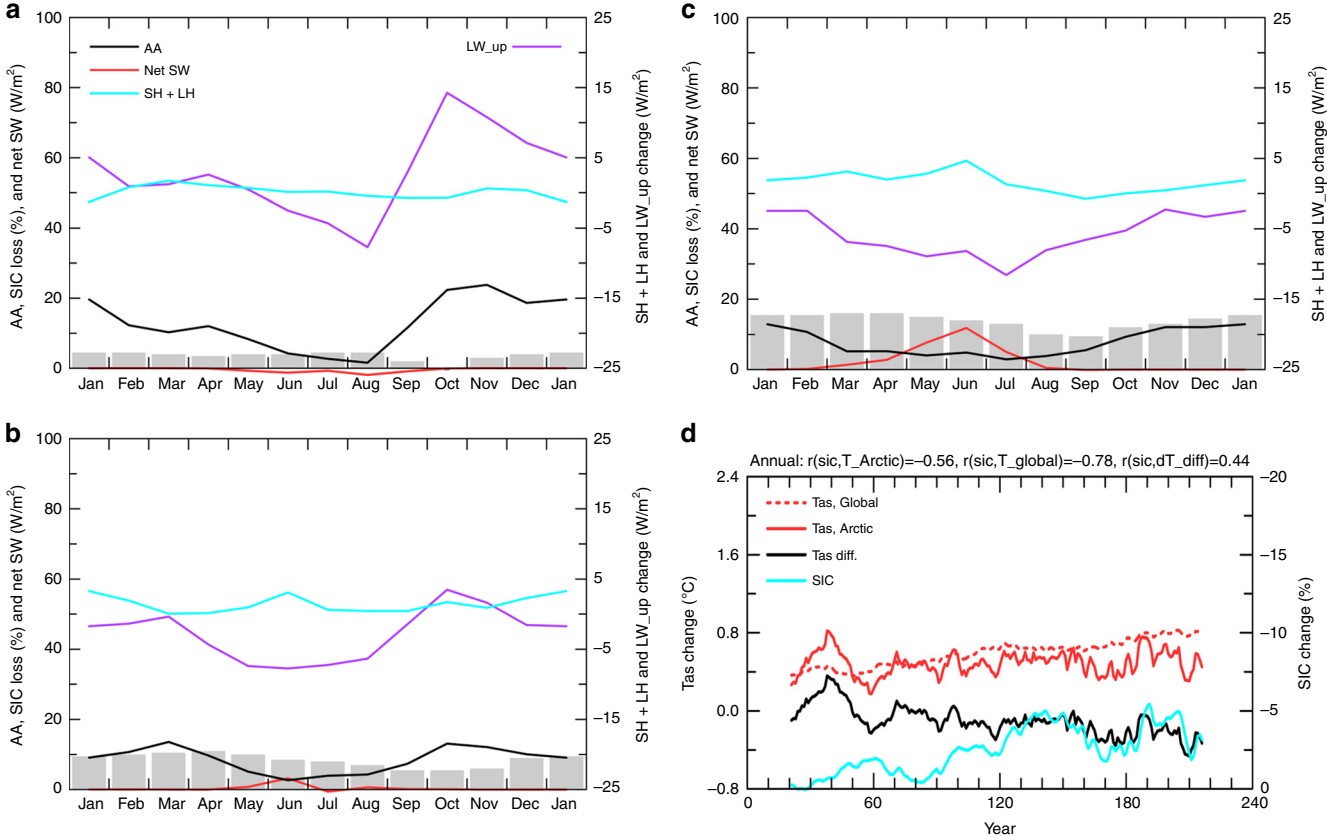

**Fig. 8** Centennial changes as a function of month from CESM1 special simulation with fixed sea-ice cover. Same as Fig. 7 except for the 1%-per-year $CO_2$ run with fixed sea-ice in calculating the ice–atmosphere and ice–ocean fluxes (FixedIce run). Note the upward longwave radiation change (LW_up) is plotted on the right y-axis in **a–c** here. **d** Time-dependent warming and sea-ice loss from CESM1 simulation with fixed sea-ice cover. Same as Fig. 3d except for the CESM1 FixedIce run

(http://cmip-pcmdi.llnl.gov/cmip5/index.html). Only nine of the models had simulations extended to 2300 under the extended RCP8.5 scenario[39]. These are bcc-csm1-1, CCSM4, CNRM-CM5, CSIRO-Mk3-6-0, GISS-E2-H, GISS-E2-R, HadGEM2-ES, IPSL-CM5A-LR, and MPI-ESM-LR. The surface warming pattern and magnitude from 1970 to 1999 and from 2070 to 2099 are comparable between the 38-model ensemble mean and the 9-model ensemble mean, suggesting that the 9-model ensemble is a reasonable representation of the larger CMIP5 model ensemble. The recent mean annual cycles of the SIC and surface energy fluxes from the 9-model ensemble (Supplementary Figure 5a) are comparable with the 38 CMIP5 models (Supplementary Figure 1b) and ERA-40 reanalysis data[47,48] (Supplementary Figure 1a). The SIC data were remapped onto a 1° grid, and all the other fields were remapped onto a 2.5° grid using a conservative mapping scheme. They were then averaged over the models (one simulation for each model) with equal weighting to create the ensemble mean, whose long-term changes were analyzed here to represent the response to future external forcing under the RCP8.5 scenario. Thus, the analyses presented here focus on the GHG-forced long-term response, not the short-term variations and decadal changes caused by internal climate variability that is a major component in recent observed changes[30,32,42]. Also, we did not examine the causes of sea-ice loss; but for the ensemble mean, the sea-ice retreat is primarily associated with GHG-induced surface warming: as the near-surface temperature increases to above the freezing point for a given month, sea ice starts to melt away (Supplementary Figure 8). Thus, the warming triggered by the increasing GHGs and enhanced by water vapor and other feedbacks (including the warming due to sea-ice loss) is the main cause of the sea-ice loss.

**CESM1 simulations.** We used the Community Earth System Model version 1 (CESM1) from NCAR[49] with the CAM4 option for its atmospheric component to make two multi-century simulations plus a 150-year pre-industrial control run. The CESM1 is a widely used fully coupled climate model that simulates the Arctic SIC and climate realistically (Supplementary Figures 1c and 2c, d). The CESM1 was run with grid spacing of 2.5° lon × ~2.0° lat for the atmospheric model, and ~1.0° lon × ~0.5° lat for the sea-ice and ocean models. The three simulations include a pre-industrial control (CTL) run with $CO_2$ fixed at 284.7 ppmv for 150 years, a standard 1% $CO_2$ run with fully coupled dynamic sea ice and a 1%-per-year increase in atmospheric $CO_2$ for 235 years reaching 10.36 times of the pre-

industrial $CO_2$ level, and a fixed sea-ice (FixedIce) run. The FixedIce run is the same as the standard 1% $CO_2$ run except that all the internally calculated ice–atmosphere, ice–ocean, and ocean–atmosphere fluxes north of 30°N were applied to the sea-ice fractional areas temporally interpolated from the monthly climatology of the CTL run, in contrast to the standard 1% $CO_2$ run in which these fluxes were applied only to the ice faction existed at the time in the model. Over a small fraction of the Arctic sea-ice area north of 30°N (mainly along the sea-ice margins at lower latitudes, Supplementary Figure 9), where sea ice melted away completely (mainly in the latter part of the simulation) and thus the ice model did not calculate these fluxes, the monthly climatology of these fluxes from the CTL run was temporally interpolated and applied to the CTL SIC fraction of these small areas, except for surface absorbed shortwave (SW) radiation which was calculated using the CTL albedo values and model-internally calculated downward SW radiation. The CTL ice-atmosphere fluxes (including LH, SH, evaporation, upward LW radiation, and surface stress) did not account for internal changes in surface temperatures and other fields, and therefore they could potentially suppress long-term changes in the Arctic if applied widely. For example, the upward LW radiation from the CTL run lacked the increasing trend associated with surface warming and therefore could weaken the Arctic warming in the lower troposphere. However, because the use of the CTL fluxes occurred only over a very small fraction of the total sea-ice area along the initial ice margins where SIC was low to start with (i.e., SIC was low in the CTL run for the small number of grid cells where the CTL fluxes were applied to the low SIC fraction; Supplementary Figure 9), the effect of this deficiency is likely to be small. The difference between the standard 1% $CO_2$ and FixedIce runs comes mainly from the application of the internally calculated fluxes over a fixed sea-ice cover from the control run in the FixedIce experiment.

In the FixedIce run, the coupler and the atmospheric and ocean components in the CESM1 only saw the fixed sea-ice cover interpolated from the CTL run north of the 30°N; the sea-ice fraction inside the sea-ice model was allowed to evolve dynamically with the fluxes returned from the coupler. Thus, the above changes made in the FixedIce run would also affect the rate of sea-ice loss through the modification to the surface fluxes.

Our main intervention of the fully coupled system in the FixedIce run is the application of the internally calculated fluxes over a fixed sea-ice cover from the CTL run (instead of only over the ice cover existed at the time inside the sea-ice

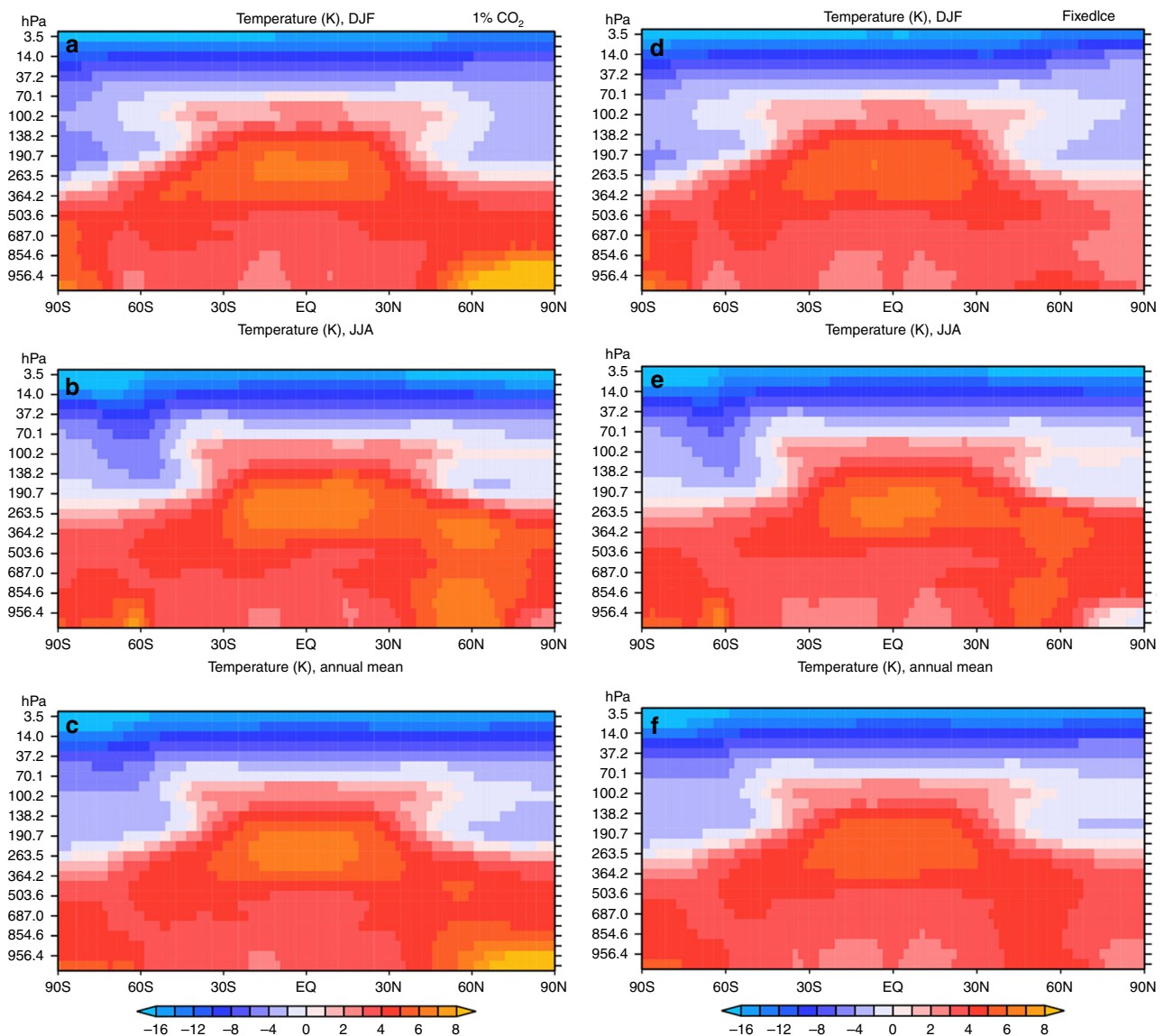

**Fig. 9** Height–latitude distributions of zonal-mean temperature change from the CESM simulations. **a-c** The standard 1% $CO_2$ run. **d-f** FixedIce run. The temperature change is relative to the control-run climatology and is around the time of the second doubling (i.e., years 131–150) of the pre-industrial $CO_2$ level. Top row: for December–January–February (DJF). Middle row: for June–July–August (JJA). Bottom row: for annual-mean. The change patterns are similar around the 1st and 3rd doubling of the pre-industrial $CO_2$

model). A secondary intervention was the use of CTL surface fluxes (instead of letting the coupler to calculate them for a water surface) over the low SIC fraction in a small number of the grid cells that initially contained sea-ice but the sea ice melted away completely. This use of the CTL fluxes was not ideal but necessary because (1) the sea-ice model did not calculate surface fluxes for grid cells without any ice, (2) the internally calculated surface temperature over these grid cells is for water surfaces and thus can't be used directly as the surface temperature for ice surfaces; and (3) the calculations of surface fluxes over sea-ice were complicated. The CTL fluxes contained only an annual cycle and were seen by the atmosphere and ocean component models; they along should not cause any long-term trends and are balanced out on an annual mean sense when applied to the small areas where ice was melted away completely (Supplementary Figure 9). That is, they should not violate the internal water and energy balance on an annual-mean basis, although they may slightly alter these balances on shorter time scales over the small number of grid cells (north of 30°N) where sea ice had melted away completely. This is much better than re-setting the ice cover to the CTL climatology after each time step, since that would provide an infinite source of ice for melting, leading to an infinite heat sink and freshwater source for the oceans. Our approach here focused on the effects on the climate (including sea ice itself) of a fixed sea-ice cover through its impact on surface fluxes; it differs from Deser et al.[37], who used an artificial LW forcing to maintain a constant sea-ice cover in a warming

experiment, and Blackport and Kushner[50], who modified ice–albedo to study the impact of sea-ice loss on the climate system.

**Top-of-atmosphere (TOA) and surface energy fluxes in the CESM1 runs**. To verify that the imposed sea-ice and flux changes to the FixedIce run did not cause major artifacts in this simulation, here we examine and compare the TOA and surface energy flux changes in the CESM1 1% $CO_2$ and FixedIce runs.

Supplementary Figure 10 shows the time series of the global-mean and Arctic-mean changes in the TOA and surface net energy fluxes from the two simulations. Over the Arctic (67°–90°N), the TOA fluxes are very similar for the two runs, while the net surface flux into the Arctic Ocean is substantially smaller in the 1% $CO_2$ run than in the FixedIce run (Supplementary Figure 10b). This is expected because the use of the fixed sea-ice concentration (SIC) in the FixedIce run reduces the upward longwave (LW) radiation and sensible and latent heat fluxes (Figs. 7, 8 and Supplementary Figures 18, 19) and therefore should lead to a larger downward net flux in the FixedIce run. Thus, the Arctic surface flux difference shown in Supplementary Figure 10b is what we should expect physically from the change we imposed, whose impact on the Arctic TOA flux is negligible. Because of this, the small difference in the global-mean TOA flux between the two runs after year ~100 (Supplementary Figure 10a) is likely due to other changes outside the Arctic region,

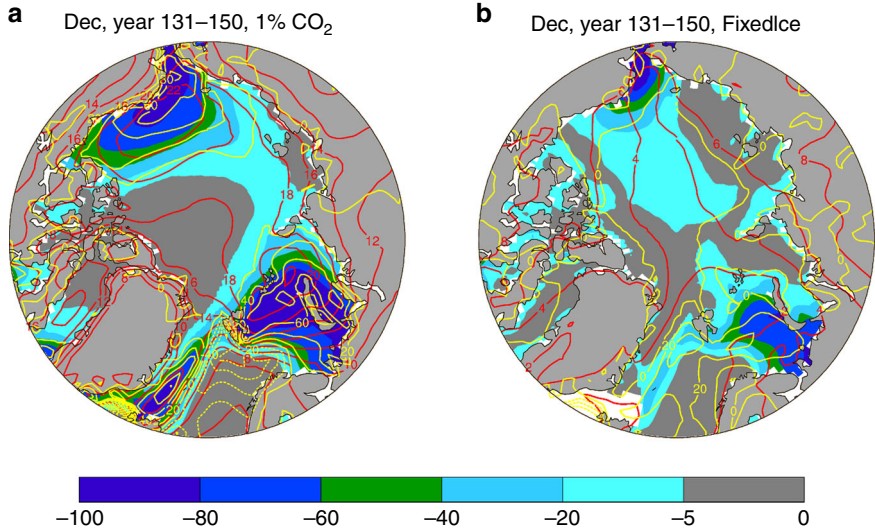

**Fig. 10** Mean changes (relative to the control-run climatology) for year 131–150. **a** from the 1% $CO_2$ run and **b** from the FixedIce run. Shown are changes in December sea-ice concentration (SIC, %, color shading), surface air temperature (Tas. °C, red contours, interval = 2), and surface latent plus sensible heat flux ((LH + SH, W m$^{-2}$, yellow contours, interval = 10, positive upward). Dashed contours are for negative values. The spatial pattern correlations are: r (SIC, Tas) = −0.20, r (SIC, LH + SH) = −0.75, and r (Tas, LH + SH) = 0.55 in **a**, and r (SIC, Tas) = 0.12, r (SIC,LH + SH) = −0.45, and r (Tas, LH + SH) = −0.28 in **b**. These correlations have a P-value below 0.01

rather than the direct effect of the imposed SIC change in the Arctic. Noticeable differences exist only after about year 100 in the global-mean fluxes between the two runs, and the TOA-minus-surface flux difference is similar between the two runs throughout the whole simulations (Supplementary Figure 10a). Thus, the global-mean and Arctic-mean TOA and surface net fluxes from the two runs are reasonable.

To examine the spatial patterns, we compare the change patterns between the two runs for the TOA and surface net energy fluxes in Supplementary Figures 11 and 12, respectively. Again, the TOA flux changes are similar in the two runs, with noticeably higher downward TOA net fluxes over most of the Arctic Ocean in the 1% $CO_2$ run (Supplementary Figure 11e) than in the FixedIce run (Supplementary Figure 11f) at the time of the 3rd $CO_2$ doubling (mainly due to increased absorption of solar radiation in the 1% $CO_2$ run), except over the areas with large SIC decreases in the FixedIce run by the 3rd $CO_2$ doubling (see Supplementary Figures 11, 13). Examination of the changes in the outgoing longwave radiation (OLR) and TOA net shortwave (SW) radiation (not shown) revealed that the increase in the Arctic TOA net flux is mainly due to the large increase in the absorbed SW radiation that exceeds the increase in OLR in the 1% $CO_2$ run, while it is mainly due to the reduced OLR (as surface upward LW radiation is reduced) as the albedo and thus SW changes are small in the FixedIce run. Thus, these TOA flux changes are expected physically based on the imposed the changes.

For the surface net energy flux (Supplementary Figure 12), the broad change patterns are similar between the two runs over most of the globe, except the Arctic where the fixed SIC in the FixedIce run is expected to reduce the absorbed SW radiation, upward LW radiation, and surface sensible (SH) and latent (LH) heat fluxes in comparison with the 1% $CO_2$ run (see Figs. 7, 8 and Supplementary Figures 18, 19). These reductions in upward fluxes lead to higher downward net energy fluxes in the FiexedIce run than in the 1% $CO_2$ over the Arctic Ocean (Supplementary Figure 12). Further examination revealed that surface downward SW radiation is reduced over the Arctic due to increased cloudiness in both simulations (Supplementary Figures 14, 15); however, due to reduced sea-ice cover in the 1% $CO_2$ run (Supplementary Figure 13), surface absorbed SW radiation is actually increased despite the decreased downward SW radiation in this run (Supplementary Figure 16a, b, e). Because of the use of ice–albedo from fixed SIC in calculating the SW flux, the reduced downward SW radiation leads to decreased absorbed SW radiation in the FixedIce run (Supplementary Figure 16b, d, f).

As the fixed sea-ice cover reduces upward LW radiation, surface net LW radiative heating increases more in the FixedIce run than in the 1% $CO_2$ run in the Arctic (Supplementary Figure 17). On the other hand, the surface LH (Supplementary Figure 18) and SH (Supplementary Figure 19) heat fluxes increase substantially only in the 1% $CO_2$ run over the Arctic, and they partially offset the large increases in the absorbed SW radiation (Supplementary Figure 16e) and the moderate increase in the net LW radiative heating (Supplementary Figure 17e) over the Arctic in the 1% $CO_2$ run, which results in only moderate changes in the net surface energy flux (Supplementary Figure 12e). This differs from the FixedIce run, in which the changes in the LH and SH fluxes are small (Supplementary Figures 17f, S18f), and the net surface energy flux change (Supplementary Figure 12f) results mainly from the reduced net SW (Supplementary Figure 16f)

and greatly increased net LW (Supplementary Figure 17f) fluxes. These changes in the FixedIce run happened over most of the Arctic, not just over the sea-ice margins (Supplementary Figure 9) where the control-run fluxes might be used. Thus, these changes are mainly due to the use of the fixed SIC in calculating the surface fluxes, rather than due to the use of the control-run fluxes over a few areas around the original sea-ice margins. This further demonstrates that the use of the fixed SIC, rather than the use of the control-run fluxes, is the main reason behind these differences between the 1% $CO_2$ and FixedIce runs. These analyses of the TOA and surface fluxes also suggest that there are no major artifacts resulting from the imposed changes in the FixedIce run.

Please note that we have provided several lines of evidence, besides using the 1% $CO_2$ minus FixedIce difference, to support our conclusion that sea-ice loss is necessary for large AA to occur. These include the diminishing AA in the 23rd century in the CMIP5 models (Fig. 3) and in the CESM1 standard 1% $CO_2$ run (Fig. 7d) when Arctic sea-ice melting becomes small. Another evidence is the strong correlation between Arctic sea-ice loss and the AA among the 38 CMIP5 models during the 21st century (Fig. 4), and the spatial and seasonal relationship between Arctic warming and sea-ice loss (and the associated surface energy flux changes). Thus, while the results from the FixedIce experiment provide a strong confirmation of the essential role of sea-ice loss, they represent only one of the several lines of evidence presented in the paper.

**ERA-Interim reanalysis data**. For the historical changes from 1979 to 2016 (Fig. 1a, b), we used the ERA-Interim reanalysis data[48], which capture the historical SIC seasonal cycle and changes fairly realistically, although the mean SIC is somewhat lower than satellite observations from NOAA (Supplementary Figure 2). The ERA-Interim data have been used to study Arctic climate and SIC change in many previous studies;[3,23,28,47] they arguably represent one of our best datasets available for the Arctic region.

**Code availability**. The code of the CESM1 model used here is available from http://www.cesm.ucar.edu/models/cesm1.2/.

## Data availability

The Methods section contains the web links to the publicly available datasets used in this study.

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

## Acknowledgements

We thank Patrick Taylor and two other anonymous reviewers for their constructive comments. We acknowledge the CMIP5 modeling groups, the Program for Climate Model Diagnosis and Intercomparison, and the WCRP's Working Group on Coupled Modelling for their efforts in making available the WCRP CMIP multi-model datasets used here. This study is partly supported by the Chinese Academy of Science Strategic Priority Research Program (Grant XDA 19070403). D.L. is also supported by the National Science Foundation of China (Grant #41430533). A.D. is supported by the National Science Foundation (AGS-1353740 and OISE-1743738), the U.S. Department of Energy's Office of Science (DE-SC0012602), and the U.S. National Oceanic and Atmospheric Administration (NA15OAR4310086). M.S. is supported by the National Key R&D Program of China (2018YFA0605901). J.L. is supported by the National Oceanic and Atmospheric Administration's Climate Program Office (NA15OAR4310163).

## Author contributions

A.D. designed the study, performed all the calculations and CESM1 simulations, made all the figures except Figure 9, which was made by Jiao Chen, and wrote the paper. D.L. participated in constructive discussions and helped improve the paper. M.S. helped modified the codes for the CESM1 experiments. J.L. contributed to the design of the CESM1 experiments.

## Additional information

**Competing interests:** The authors declare no competing interests.

