## [Peer Review File · Nature Communications]

Reviewer #1 (Remarks to the Author):

Review of NCOMMS-17-03941-T "Arctic amplification caused by sea-ice loss under future warming"

Overall:

Arctic Amplification (AA) is a fundamental aspect of the Earth's response to anthropogenic climate change, a feature known for more than 40 years. However, intense debate continues, as to the cause. This manuscript presents an analysis of long, 400-year climate change simulations under RCP8.5 forcing to assess the influence of sea ice loss on the magnitude of Arctic amplification. In my view, using long, 400-year climate simulations are a novel and a useful perspective not highlighted previously. However, I find it difficult to assess the claims made in this manuscript from the analysis presented. In my view, the main conclusions drawn that "sea-ice is the primary reason for the existence of large AA" and that "other processes can only modulate the sea-ice loss-induced amplification but they cannot cause large AA without sea ice loss" cannot be made from the presented analysis. I agree that sea ice loss and the associated responses of surface turbulent fluxes and seasonal ocean heat storage are incredibly important aspects of AA and I see value in the analysis itself. But, the bold assertions made within this manuscript are not supported within the manuscript. I see value in the analysis and recommend major revision of this manuscript changing the tone and scope of the main conclusions.

Major comments:

My biggest concern is that the authors neglect the fact that by the 23rd century the Arctic and global climate state differs significantly from 1970-1999 and even 2070-2099. No reference is made to these differences in the climate state. However, a lot of changes have occurred by the 22nd century and I would argue we are no longer looking at the same climate state. Atmosphere and ocean circulations, clouds, temperatures, and lower tropospheric stability have changed; feedbacks and interactions between climate system components have likely taken a different shape. For instance, as the Arctic likely becomes >10 K warmer by the 22nd century the negative Planck feedback becomes notably stronger in fall and winter. As sea ice disappears there will also be significant changes to the lower tropospheric temperature, humidity, clouds, and atmospheric and oceanic circulations that will change the interactions and may change the character of further amplification. Additionally, throughout the manuscript it seems to be continually forgotten that the Arctic temperatures have already amplified and remain amplified relative to the historical period (1970-1999) in the 23rd century. The fact that the climate state itself changes between the 21st and 23rd century indicates to me that the presented statistical analysis cannot be used to state that sea ice is the primary reason for the existence of large AA. The accurate argument based on the analysis

presented seems to be that no further Arctic amplification occurs after sea ice disappears. This is very different than saying sea ice causes Arctic Amplification.

The climate system is complex web of interacting components that often compensate for one another. Thus, given an anthropogenic forcing this analysis method does not test whether AA would have occurred in the absence of sea ice loss. A more direct test of this hypothesis that large AA cannot occur without sea ice loss could be done by performing model simulations holding sea ice constant under and RCP8.5 forcing.

Other explanations for the slowing of AA exists besides sea ice loss. It is possible AA slows by the 23rd century as a result of non-sea ice feedbacks and processes that limit the overall warming especially in fall in winter. Also, ocean circulation-related feedback processes not expected to be evident during 21st will begin to take shape over a couple centuries. In other words, the system may not be capable of sustaining “runaway Arctic warming.”

Additionally, the manner in which this paper is laid out and references to figures (both inside the manuscript and supplemental figures) is very difficult to follow. I found it dizzying, in fact. Also, the supplementary figures contain too much information making it very difficult to glean the information that the authors are trying to express.

Lastly, there is no discussion of the fact that the RCP8.5 forcing declines after 2100, which could contribute to the slowing of AA.

Minor Comments:

Line 30: I think you mean “warm” not “warming”

Line 76-77: It doesn't seem to me that the claim ‘the largest release of LH and SH fluxes is delayed to January-February by the end of the 22nd century’ made here is supported by Fig. S2. It seems to me that the largest negative LH fluxes (surface warming atmospheric) occur in June/July between 1970-1999, in October/November in 2070-2099, and in November/December in 2170-2199 and 2270-2299. It appears you may be mixing up changes in LH and SH fluxes with the absolute value of the mean annual cycles (Also there is a typo in the S2 figure caption, ‘22999’ should be ‘2299’).

Line 84: “and so do” should be replaced with “as do”

Line 85: It is important here to be clear that you are talking about continued warming. These seasons are still significantly amplified from the 1970-1999 period.

Line 95-96: replace “left to be melt” with “left to be melted” or “left to melt”

Line 129: I disagree with the use of ‘strongly’

Signed: Patrick Taylor

Reviewer #2 (Remarks to the Author):

General:

Pardon the pun, but Arctic amplification, hereafter AA (usually defined as the outsized rise in near surface air temperatures over the Arctic compared to the trend for the globe as a whole), has become a very hot topic. While the literature is filled with studies looking at how AA, through altering horizontal temperature gradients, may (or may not) influence weather patterns both beyond and within the Arctic, there is also considerable controversy on the fundamental causes of AA itself.

There is strong evidence of a central role of sea ice loss – as the climate warms, areas of dark open water form earlier in the melt season, readily absorbing solar energy. When the sun sets in autumn, the extra energy stored in the ocean mixed layer is released upwards via strong turbulent fluxes and longwave radiation. This can be seen in the correspondence between areas of ice loss and local maxima in temperature trends, and in the seasonality of the temperature changes – biggest in autumn and winter, and small in summer. There is also evidence for roles of increased cloud cover and water vapor (both increasing the downward longwave flux), the strong stability of the near surface atmosphere in the Arctic, which limits vertical mixing, and the so-called Planck effect. Other recent studies emphasize that AA is primarily an advection phenomenon – changes in atmospheric

circulation lead to AA either directly through poleward energy transport or through the effects of more water vapor and cloud cover increasing the downward longwave radiation flux. For example, recent work by Dr. Sukyoung Lee (Penn State) points to heating in the tropical western Pacific forcing poleward amplification of Rossby waves which then warms the Arctic. The advection point of view bears strongly on arguments surrounding whether AA can influence mid-latitude weather patterns – if AA is more driven by advection from lower latitudes than by ice loss, then the argument that AA can lead to changes in mid-latitude weather patterns is basically wrong – it is, in effect, putting the cart before the horse.

Having followed AA debate fairly closely, I side with the authors of the present paper that sea ice loss is a key driver. The authors have focused a suite of simulations from the CMIP-5 archive to see how AA evolves in relation to sea ice loss, and I like the idea of looking as far out as the end of the 22nd century to see what happens to AA as the Arctic loses even its winter sea ice. I'm not sure if anyone else has done this. The argument that AA goes away once the sea ice is essentially gone is compelling.

However, the challenge that I see in getting this paper published in "Nature Communications" in its present form is that that authors need to do a better job in distilling their arguments into a more focused and digestible form. Many of the figures are much too busy to effectively interpret, and several, such as Figure 4, are indecipherable. This paper could have a strong impact, but to do so, there needs to be a much cleaner presentation that focuses on the essentials.

Along the lines of interpretation, I am wondering why the authors fail to address the upward longwave flux from the surface. In my view, the net longwave flux, such as plotted in Figure 2 and Figures S2 and S3, is not really the issue, for if there is more upward emission from the surface there will also be more balancing downward emission, especially if one is also putting more water vapor into the air from the latent heat flux. What I think one wants to look at is the heat fluxes INTO the atmosphere, and the turbulent sensible and latent heat fluxes don't tell the whole story. Assuming blackbody radiation, at a reasonable ice surface temperature of -20 deg. C, there is an upward flux of 232 W m⁻², compared to an open water value at +2 deg. C of 325 W m⁻², a difference of 93 W m⁻², which is a pretty big number (assuming I have my calculations correct). I bring this up especially because the changes in the turbulent heat fluxes into the atmosphere such as shown in Figure 2 don't look to be very big by themselves (see comments below).

Another issue is that authors could do a better job at discussing their own conclusions of a central role of sea ice loss on AA with ideas presented in other studies. The view of the authors, as discussed on page 5, is that that "other" processes can only modulate the AA induced by sea ice loss but cannot cause large AA without sea ice loss. However, the way I see it is that we are dealing with a highly coupled set of processes. Yes, I agree that sea ice loss is central to AA, but why the sea ice loss in the first place? Clearly, as the authors discuss on page 6 and with reference to Figure S6, it is

because it is warming and will continue to do so in the future under GHG forcing. However, changes in atmospheric circulation bringing more energy into the Arctic from lower latitudes along with more cloud cover and water vapor, while bolstering the warming, also leads to less sea ice, which then promotes enhanced upward surface fluxes that warm the atmosphere. Along similar lines, extensive open water in the Barents Sea seen in recent winters is expressed as a prominent peak in temperature anomalies, but the ice loss itself seems at least partly related to enhanced inflow of Atlantic derived waters. The point here is that ice loss leads to Arctic amplification, but processes in addition to the direct effects of GHG warming affect the sea ice loss. The argument that the “other” processes can only modulate the sea ice loss induced amplification but cannot cause large AA without sea ice loss is hence not quite right in my opinion.

Specific:

Abstract, sentence 1: Be careful with loose terminology. Exactly how are the authors defining Arctic Amplification? Is it the Arctic temperature trend as compared to the global average trend? There are parts of the “rest of the world”, notably around the Antarctic Peninsula, that show a warming trend as large as in the Arctic. And warming over what period, the instrumental record?

Line 32: For context, it should be noted here that the observations show downward trends in ice extent, strongest at the end of the melt season in September. The way the text is written might give the unintended impression that ice loss is something out there in the future.

Line 43: It should be mentioned right here that the study uses simulations from the CMIP-5 archive. Don't leave the reader wondering.

Line 45: It might be worth mentioning here that (as I understand it) current emission rates are pretty much in line with RCP 8.5, hence it is not an extreme scenario.

Line 46: Be specific with terminology. By saying “ice loss peaks in the latter half of the 21st century” are you saying that the annual rate of change is largest in that period?

Figure 1: This figure might be much easier to digest if presented as bar plots. For example, for each month, show three bars, of different colors, side by side: the SIC change, Arctic temperature change and AA.

Figure 1a: In the text, briefly discuss why we see the peak in SIC change in July.

Figure 2: I'm a bit puzzled by this figure. Figure 1 shows AA in terms of the ratio between the Arctic and global average temperature, while Figure 2 (and others that follow) show instead the difference between the global and Arctic temperature. I find this to be confusing. Also, trying to show the temperature difference on the same scale as the energy fluxes is not effective - for example, on the top panel, the energy fluxes range from -30 to +30 W m⁻² while the range for the temperature difference is only from about zero to plus 1.5 so it's really hard to see what is going on. A better presentation is warranted.

Figure 2: For the top panel, the results seem to show that the difference between 2070-2099 and 1970-1999 in the turbulent sensible heat flux peaks at less than 10 W m⁻²; the peak difference for the turbulent latent heat flux is only around 10 W m⁻². These are small values, and would seem to counter the argument that AA is related to enhanced surface heat fluxes associated with sea ice loss. Why are these fluxes so small? Or am I simply missing something?

Figure 2 and elsewhere: As discussed above, I don't think that the net longwave flux is the thing that the authors should be focusing on – the upward longwave flux strikes me as much more relevant as it is a flux of energy into the lowest layer of the atmosphere from the surface.

Figure 3 caption: Should this instead be “40 year periods centered on the plotted year”?

Figure 4: This figure is entirely undecipherable – way too many overlapping lines. It will be much better to show individual panels for the different variables.

Throughout the text: Correlations are cited without corresponding p-values.

Reviewer #3 (Remarks to the Author):

The paper contrasts Arctic warming and Arctic amplification in periods with substantial sea ice retreat and after the loss of most of the initial sea ice area to determine the role of sea ice and sea

ice loss for Arctic amplification. It thus addresses an important problem in Arctic climate research with an interesting and novel idea. Unfortunately, the specific approach taken here may suffer from a fundamental flaw: By contrasting AA in the 21st and 22nd century (with sea ice loss) and the 23rd century (without sea ice loss), the authors compare episodes of global warming that are very different in terms of the radiative forcing applied and the global warming rate. The manuscript does not contain sufficient details on these issues, but they certainly carry the risk to mask the signal the analysis is trying to reveal.

The arguments not affected by these issues, such as temporal and spatial correlations between Arctic warming and sea ice loss, are by themselves not strong enough to support the papers' conclusions. I therefore suggest to repeat the analysis in experiments with abrupt or steady forcing, and make sure that periods with and without sea ice are comparable in terms of the global climate change signal.

Response to Reviewers' comments for manuscript NCOMMS-17-03941-T

We thank the reviewers for their constructive comments. We have performed additional multi-century simulations using the NCAR CESM1 to address the concerns from Reviewer #1 and #3 and to further quantify the effects of sea-ice loss on Arctic Amplification (AA). We apologize for the long delay in providing a revised version, mainly because the time needed for us to learn the CESM1 codes and its ice model in particular in order to make the appropriate code changes for the experiments needed to strengthen this study. The design of the experiments also changed a few times, and each time we needed to make new coding changes. All these have delayed the revision of the paper, but we are happy that we are able to carry out the experiments needed to significantly strengthen this manuscript. We hope the new evidence will help convince the reviewers that AA is indeed caused mainly by sea-ice loss and that no significant AA would occur if there is no sea-ice loss. We do not claim that longwave (LW) radiation does not play a role in Arctic (and global) warming, but our results strongly suggest that LW radiation cannot cause an amplified Arctic warming compared with the rest of the world if there is no sea-ice loss. In addition, any external forcing (e.g., CO₂ increases) or internal processes (e.g., water vapor feedback, poleward energy transport, or increased LW heating due to increased clouds or water vapor) can indirectly contribute to AA by causing sea-ice loss in the Arctic.

Reviewer #1 (Remarks to the Author):

Review of NCOMMS-17-03941-T "Arctic amplification caused by sea-ice loss under future warming"

Overall:

Arctic Amplification (AA) is a fundamental aspect of the Earth's response to anthropogenic climate change, a feature known for more than 40 years. However, intense debate continues, as to the cause. This manuscript presents an analysis of long, 400-year climate change simulations under RCP8.5 forcing to assess the influence of sea ice loss on the magnitude of Arctic amplification. In my view, using long, 400-year climate simulations are a novel and a useful perspective not highlighted previously. However, I find it difficult to assess the claims made in this manuscript from the analysis presented. In my view, the main conclusions drawn that "sea-ice is the primary reason for the existence of large AA" and that "other processes can only modulate the sea-ice loss-induced amplification but they cannot cause large AA without sea ice loss" cannot be made from the presented analysis. I agree that sea ice loss and the associated responses of surface turbulent fluxes and seasonal ocean heat storage are incredibly important aspects of AA and I see value in the analysis itself. But, the bold assertions made within this manuscript are not supported within the manuscript. I see value in the analysis and recommend major revision of this manuscript changing the tone and scope of the main conclusions.

Response:

We present the following evidence for the central role of sea-ice loss for AA:

1. Large AA exists only when there is significant sea-ice loss and the AA diminishes when sea-ice melts away in the extended CMIP5 model simulations during all periods (not just in the 23rd century);
2. The amplified Arctic warming occurs only during the cold season and only over the areas with large sea-ice loss. The sea-ice related mechanism can easily explain these seasonal and spatial patterns of AA, while it is hard for all the other mechanisms to account for these features;
3. Models with larger sea-ice loss produce larger AA among the CMIP5 models. This suggests a major role of sea-ice loss, and this cannot be explained easily by the other mechanisms;
4. In our new CESM1 simulations, using a constant Arctic sea-ice cover in calculating surface fluxes resulted in no significant AA under a 1%/yr CO₂ increase throughout the simulation. This strongly suggests that without the impact of sea-ice loss on surface exchange fluxes, all other mechanisms cannot produce significant AA. Other processes, such as increased longwave (LW) radiation from increased CO₂ and water vapor, can contribute to Arctic (and global) warming, but it cannot produce the amplified warming seen only over the areas with sea-ice loss.
5. We showed that the amplified warming occurs only over the cold winter months and only over areas with sea-ice loss due to the release of SH and LH fluxes during these months over the newly opened waters. The extra LH fluxes increase the LW heating while the extra SH fluxes directly warms the lower troposphere, both contributing a warmer surface. Thus, increased downward LW radiation is also involved locally, but this is caused by the LH release from the newly opened waters, not due to transport from lower latitudes (which would not be concentrated over the areas with large sea-ice loss). The latter can contribute to the overall warming in the Arctic that helps it in pace with the global warming rate (but resulting in no significant AA, as shown in our fixed SIC CESM1 run).

Major comments:

My biggest concern is that the authors neglect the fact that by the 23rd century the Arctic and global climate state differs significantly from 1970-1999 and even 2070-2099. No reference is made to these differences in the climate state. However, a lot of changes have occurred by the 22nd century and I would argue we are no longer looking at the same climate state. Atmosphere and ocean circulations, clouds, temperatures, and lower tropospheric stability have changed; feedbacks and interactions between climate system components have likely taken a different shape. For instance, as the Arctic likely becomes >10 K warmer by the 22nd century the negative Planck feedback becomes notably stronger in fall and winter. As sea ice disappears there will also be significant changes to the lower tropospheric temperature, humidity, clouds, and atmospheric and oceanic circulations that will change the interactions and may change the character of further amplification.

Response: While the CMIP5 model simulated climate by the 23rd century may differ substantially from the current climate and that might become a factor for AA (we now point out this on p.8), it should be noted that the seasonal, spatial, and temporal dependence of AA on sea-ice loss is seen in all the time periods, not just in the 23rd century, and the AA vs. SIC loss relationship among CMIP5 models (Fig. 4 in revised version) is for the 21st century. The 23rd century is used because the SIC is

low and thus sea-ice loss is low by that time in the CMIP5 simulations. We have now added the CESM1 simulations in which atmospheric CO₂ increases by 1%/yr from the pre-industrial level, thus atmospheric CO₂ level and the mean climate during its first 100 years or so are not very different from our recent and near-future climate; yet we show that the AA disappears during this period when a constant SIC was used in calculating the surface fluxes (see Fig. 6b and Fig. 8a). Furthermore, no significant AA is seen throughout the 235 years of simulation (Fig. 6b), over which the CO₂ level varies from 284.7ppmb to over 2847 ppmv and thus the mean climate state changes a lot (see global T_{as} change in Fig. 6b). This further suggests that the AA is insensitive to the mean climate state if SIC is fixed in calculating the surface fluxes. Thus, we do not think the future mean climate state plays an important role in producing the AA vs. sea-ice loss relationship in the extended CMIP5 simulations.

Additionally, throughout the manuscript it seems to be continually forgotten that the Arctic temperatures have already amplified and remain amplified relative to the historical period (1970-1999) in the 23rd century. The fact that the climate state itself changes between the 21st and 23rd century indicates to me that the presented statistical analysis cannot be used to state that sea ice is the primary reason for the existence of large AA. The accurate argument based on the analysis presented seems to be that no further Arctic amplification occurs after sea ice disappears. This is very different than saying sea ice causes Arctic Amplification.

Response: Please note that we focused on the centennial changes in order to examine the relationship between the AA and sea-ice loss. It is true that the Arctic T_{as} change from the present (1970-1999) to the 23rd is amplified relative to global-mean T_{as} change, but this is due the amplification occurred during the 21st and 22nd centuries when sea-ice loss existed. See our response above regarding the potential effect of the mean climate state. Please note that we presented various evidence (including the seasonal and spatial arguments) besides the lack of AA in the 23rd century when little sea ice exists. When sea ice melts away, the AA disappears. That itself is an important result. Combined with our other evidence that shows the mean state is not a factor, it strongly suggests that sea-ice loss is necessary for large AA to occur and that all the other mechanisms cannot produce large AA without sea-ice loss. This is in addition to our arguments regarding the seasonal, spatial, temporal (i.e., Fig. 4d and Fig. 7d), and inter-model (Fig. 4) dependence of the AA on sea-ice loss. The new CESM1 simulations further confirm the crucial role of sea-ice loss directly (i.e., without its impact on surface fluxes, AA won't occur. This implies that all the other mechanisms can't produce AA without sea-ice loss).

The climate system is complex web of interacting components that often compensate for one another. Thus, given an anthropogenic forcing this analysis method does not test whether AA would have occurred in the absence of sea ice loss. A more direct test of this hypothesis that large AA cannot occur without sea ice loss could be done by performing model simulations holding sea ice constant under and RCP8.5 forcing.

Response: Following this suggestion, we designed and performed two multi-century simulations using the NCAR CESM1, a fully coupled climate model: A standard 1%/yr CO₂ increase run from

the pre-industrial level and another similar run except that fixed (but monthly-varying) sea-ice cover (based on control climatology) was used in calculating the surface fluxes (see Methods). This approach only alters the calculation method (sea-ice vs. ocean water) for the surface fluxes. It does not violate the energy or mass conservations inside the CESM1, thus it is better than artificially re-setting the sea-ice cover to the control climatology because the latter would provide an unlimited source of sea ice for melting in the model (and thus lead to an unlimited heat sink and freshwater source that would cause artificial long-term changes). Our approach does not alter the sea-ice volume inside the model, it only changes the way the surface fluxes are calculated, i.e., whether they are calculated over an ocean water surface or over a sea-ice surface. The 1%/yr CO₂ increase is a much simpler forcing scenario to implement and interpret than the time-varying RCP8.5 forcing.

Other explanations for the slowing of AA exists besides sea ice loss. It is possible AA slows by the 23rd century as a result of non-sea ice feedbacks and processes that limit the overall warming especially in fall in winter. Also, ocean circulation-related feedback processes not expected to be evident during 21st will begin to take shape over a couple centuries. In other words, the system may not be capable of sustaining “runaway Arctic warming.”

Response: There are countless possibilities that could potentially cause AA, as mentioned in your comments. Clearly one cannot investigate each of them one-by-one in a single study to quantify their role. Here, we focused on the role of sea-ice loss, and by comparing the cases with and without sea-ice loss, we can quantify the role of sea-ice loss, and infer the role of the combined effect from all the other possibilities. This is a fairly standard approach in climate modeling studies.

Again, please note that we presented much more evidence than just the changes in the 23rd century, and have now showed that the mean climate state, at least according to our CESM1 simulations, is not a significant factor for the AA under rising CO₂ from 284.7 to over 2847ppm. We now emphasize the seasonal, spatial, temporal, and inter-model dependence of the AA on sea-ice loss, besides the new CESM1 simulations. Please note that large amplification of surface warming occurs only during the cold season and only over the areas with large sea-ice loss and large SH+LH release. We could not think of any other mechanisms that can easily account for these seasonal and spatial features, as well as the inter-model dependence. The new CESM1 simulations further directly confirm that it is necessary to have sea-ice loss in order to have large AA because of the associated changes in surface fluxes. We think it is important to examine the seasonal, spatial, temporal, and inter-model characteristics of the AA to help understand its causes, not just through an apparent role (of LW and other fluxes) in Arctic-mean and annual warming, as such a warming effect exists all the time in the current climate and it may or may not cause an amplification of surface warming relative to global warming under increasing CO₂.

More specifically, increased CO₂ and water vapor will increase downward longwave radiation that causes the warming in the Arctic and over the globe in the first place; thus a role of increased downward LW radiation in surface warming is expected over the Arctic and all other regions, but this role does not necessarily lead to an AA. Therefore, all the studies that show a role of increased downward LW radiation in Arctic warming do not necessarily imply a role of LW radiation in

producing the AA, that is, the LW-induced warming could be similar for the Arctic and for the globe. One needs to look at where and when the amplification occurs in the Arctic under rising CO₂, and how such a spatial and seasonal patterns of the AA can be explained by a potential mechanism.

Another example is the poleward transport of sensible and latent energy, which happens all the time in our current climate that is required to balance Arctic LW cooling to maintain current meridional temperature gradient. In the future warmer climate, such a poleward transport of energy may change associated with a changed meridional temperature gradient (and thus circulation), but whether such a change in the meridional energy transport would contribute to the AA is a very different story, because such a change may be needed just to balance the altered LW cooling the Arctic. Our results suggest that any such change would not lead to large AA (as shown in our CESM1 run with fixed SIC for flux calculations), which occurs mainly over areas with large sea-ice loss during the cold season. It would be difficult to image that an increased poleward energy transport would cause an AA only over areas and only over months with large sea-ice loss during the cold season, and then the AA would disappear when a fixed SIC was used in calculating the surface fluxes.

Additionally, the manner in which this paper is laid out and references to figures (both inside the manuscript and supplemental figures) is very difficult to follow. I found it dizzying, in fact. Also, the supplementary figures contain too much information making it very difficult to glean the information that the authors are trying to express.

Response: We took advantage of the longer page limit of *Nature Communications* in the revised version and re-organized the text and figures substantially. We hope it is easier to read than previously. Even so, we could not describe the key features in the figures in the way we would like for *J. Climate* or *Climate Dynamics*. Here we can only cite the figures to support the arguments we're trying make. This is common for all the short articles in Nature or Science or their specialty journals.

Lastly, there is no discussion of the fact that the RCP8.5 forcing declines after 2100, which could contribute to the slowing of AA.

Response: We think the following text (lines 46-49 on p2 in the 1st submission and lines 55-58 on p3 in the revised version) already pointed out this fact of declining forcing after 2100:

“In the CMIP5 simulations, Arctic sea-ice loss peaks in the latter half of the 21st century; thereafter, both Arctic sea-ice cover (SIC) and sea-ice loss diminish while global warming continues, albeit at a slower pace after the late 22nd century due to reduced GHG forcing³⁰ (Fig. S1 in Suppl. Information).” Again, we don't think this change in forcing is a factor, as we examined the AA.vs. SIC relationship throughout the simulations in both CMIP5 models and in CESM1 experiments.

Minor Comments:

Line 30: I think you mean “warm” not “warming”

Corrected. Thanks.

Line 76-77: It doesn't seem to me that the claim 'the largest release of LH and SH fluxes is delayed to January-February by the end of the 22nd century' made here is supported by Fig. S2. It seems to me that the largest negative LH fluxes (surface warming atmospheric) occur in June/July between 1970-1999, in October/November in 2070-2099, and in November/December in 2170-2199 and 2270-2299. It appears you may be mixing up changes in LH and SH fluxes with the absolute value of the mean annual cycles (Also there is a typo in the S2 figure caption, '22999' should be '2299').

Please note the cited figure for that sentence was Fig.2a-b, which shows the centennial change in the LH and SH. Fig. S2 only shows the mean annual cycle for different time periods, not the change. The sentence is revised on p.5 to "The largest release of the extra LH and SH is delayed to January-February by the end of 22nd century from November-December in the 21st century as the maximum sea-ice loss moves to latter months (Fig. 3a-b)."

The typo in Fig.S2 caption is corrected. Thanks.

Line 84: "and so do" should be replaced with "as do"

Done.

Line 85: It is important here to be clear that you are talking about continued warming. These seasons are still significantly amplified from the 1970-1999 period.

The sentence is revised to "Without the extra heating from the ocean (on top of the mean seasonal cycle, Fig. S4c-d), the Arctic warming during the 23rd century shows little seasonal variation, ..."

Line 95-96: replace "left to be melt" with "left to be melted" or "left to melt"

Changed to "left to melt"

Line 129: I disagree with the use of 'strongly'

We hope that the addition of the results from the new CESM1 simulations has greatly strengthened statement.

Signed: Patrick Taylor

Thank you so much, Patrick, for your constructive comments, especially for the suggestion to perform additional simulations. They motivated us to perform the new CESM1 experiments, which,

in our opinion, provide direct evidence that sea-ice loss is necessary for large AA to occur. We hope the revised version has addressed most of your concerns.

Reviewer #2 (Remarks to the Author):

We thank the reviewer for the many constructive comments and positive view of the manuscript.

General:

Pardon the pun, but Arctic amplification, hereafter AA (usually defined as the outsized rise in near surface air temperatures over the Arctic compared to the trend for the globe as a whole), has become a very hot topic. While the literature is filled with studies looking at how AA, through altering horizontal temperature gradients, may (or may not) influence weather patterns both beyond and within the Arctic, there is also considerable controversy on the fundamental causes of AA itself.

There is strong evidence of a central role of sea ice loss – as the climate warms, areas of dark open water form earlier in the melt season, readily absorbing solar energy. When the sun sets in autumn, the extra energy stored in the ocean mixed layer is released upwards via strong turbulent fluxes and longwave radiation. This can be seen in the correspondence between areas of ice loss and local maxima in temperature trends, and in the seasonality of the temperature changes – biggest in autumn and winter, and small in summer. There is also evidence for roles of increased cloud cover and water vapor (both increasing the downward longwave flux), the strong stability of the near surface atmosphere in the Arctic, which limits vertical mixing, and the so-called Planck effect. Other recent studies emphasize that AA is primarily an advection phenomenon – changes in atmospheric circulation lead to AA either directly through poleward energy transport or through the effects of more water vapor and cloud cover increasing the downward longwave radiation flux. For example, recent work by Dr. Sukeyoung Lee (Penn State) points to heating in the tropical western Pacific forcing poleward amplification of Rossby waves which then warms the Arctic. The advection point of view bears strongly on arguments surrounding whether AA can influence mid-latitude weather patterns – if AA is more driven by advection from lower latitudes than by ice loss, then the argument that AA can lead to changes in mid-latitude weather patterns is basically wrong – it is, in effect, putting the cart before the horse.

Response: These were the exact motivation for this study. As pointed out in the paper and in our response to Reviewer #1's comments, one needs to examine the seasonal and spatial characteristics of the AA more closely to gain some insights into its possible cause(s). Poleward transport of sensible and latent energy occurs all the time in current climate that is required to balance the large longwave (LW) cooling in the Arctic, and increased downward LW radiation from increased CO₂ and water vapor is expected to contribute to surface warming in the Arctic and most of the globe, as this is the cause of global warming in the first place. However, these roles in maintaining a warmer Arctic air temperature (as shown in Gong et al. 2017, ref. 34 of revised version) than the case without these effects do not necessarily imply a contribution to AA, because they do not necessarily produce a warming difference between the Arctic and the rest of the world. The fact that Arctic warming is similar to the global warming rate during the warm season (when increased downward LW radiation also exists due to increased CO₂ and water vapor) is a good example of that. Besides the seasonality issue, the spatial co-location of the amplified warming with the sea-ice loss also clearly suggests a crucial role of sea-ice loss for AA, as all the other mechanisms cannot easily

produce such a pattern without involvements of sea ice. These basic considerations motivated us to quantify the role of sea-ice loss further using model simulations.

Having followed AA debate fairly closely, I side with the authors of the present paper that sea ice loss is a key driver. The authors have focused a suite of simulations from the CMIP-5 archive to see how AA evolves in relation to sea ice loss, and I like the idea of looking as far out as the end of the 22nd century to see what happens to AA as the Arctic loses even its winter sea ice. I'm not sure if anyone else has done this. The argument that AA goes away once the sea ice is essentially gone is compelling.

Response: Thanks for the positive view of our study. That was the exact reason for us to extend to the 23rd century because sea ice would no longer exist over much the Arctic and that provides a good contrasting case. However, Reviewer #1 was concerned that the mean climate state may be very different by the 23rd century and that could become a factor for the AA-ice relationship. In response to this criticism, we point out that the seasonal, spatial and inter-model dependence of the AA on sea-ice loss is seen throughout the simulations, not just during the 23rd century, and we have now added new results from the new CESM1 experiments with and with sea-ice loss.

However, the challenge that I see in getting this paper published in "Nature Communications" in its present form is that that authors need to do a better job in distilling their arguments into a more focused and digestible form. Many of the figures are much too busy to effectively interpret, and several, such as Figure 4, are indecipherable. This paper could have a strong impact, but to do so, there needs to be a much cleaner presentation that focuses on the essentials.

Response: Thanks for the suggestion. We have taken the advantage of *Nature Communication's* longer page limit to re-organize and expand the text substantially. The figures are also re-drawn with fewer lines in many cases and more of them are put into the main paper instead of the Suppl. Information. The old Fig. 4 (new Fig. 5) is revised by removing contours for the net energy flux (not essential), so it is easier to read. We also removed the net energy and net LW fluxes from the flux change Figures and combined the SH and LH into one line. This makes them much easier to read, despite that we added a new line for upward longwave flux into the Figures.

Following Reviewer #1's suggestion, we have performed additional modeling experiments with and without sea-ice loss under a constant CO₂ forcing (1%/yr increase) to further quantify the role of sea-ice loss. The new results significantly strengthen our arguments that it is necessary to have sea-ice loss in order for AA to occur and that all the other mechanisms can't produce large AA without sea-ice loss.

Along the lines of interpretation, I am wondering why the authors fail to address the upward longwave flux from the surface. In my view, the net longwave flux, such as plotted in Figure 2 and Figures S2 and S3, is not really the issue, for if there is more upward emission from the surface there will also be more balancing downward emission, especially if one is also putting more water vapor into the air from the latent heat flux. What I think one wants to look at is the heat fluxes INTO the atmosphere, and the turbulent sensible and latent heat fluxes don't tell the whole story. Assuming blackbody radiation, at a reasonable ice surface temperature of -20 deg. C, there is an upward flux of 232 W m⁻², compared to an open water value at +2 deg. C of 325 W m⁻², a difference of 93 W m⁻², which is a pretty big number (assuming I have my

calculations correct). I bring this up especially because the changes in the turbulent heat fluxes into the atmosphere such as shown in Figure 2 don't look to be very big by themselves (see comments below).

Response: Thanks for this constructive comment. We agree that the upward LW flux is of interest here, and we have added it to many of the Figures in the revised version. It does show large increases during the cold season due to the enhanced surface warming during these cold months. As the sea ice disappears and the warming becomes uniform through the year in the 23rd century, the upward LW flux increases uniformly throughout the year at a lower pace (Fig. 3c). Clearly, whether the surface is covered by sea ice during the cold season makes a huge difference for the upward LW, SH and fluxes, and they all contribute to the enhanced warming of the air near the surface and in the lower troposphere. This is the key point behind our argument of the central role of sea-ice loss for AA. We have revised the manuscript accordingly to include the upward LW flux in the discussion through the paper.

Another issue is that authors could do a better job at discussing their own conclusions of a central role of sea ice loss on AA with ideas presented in other studies. The view of the authors, as discussed on page 5, is that that "other" processes can only modulate the AA induced by sea ice loss but cannot cause large AA without sea ice loss. However, the way I see it is that we are dealing with a highly coupled set of processes. Yes, I agree that sea ice loss is central to AA, but why the sea ice loss in the first place? Clearly, as the authors discuss on page 6 and with reference to Figure S6, it is because it is warming and will continue to do so in the future under GHG forcing. However, changes in atmospheric circulation bringing more energy into the Arctic from lower latitudes along with more cloud cover and water vapor, while bolstering the warming, also leads to less sea ice, which then promotes enhanced upward surface fluxes that warm the atmosphere. Along similar lines, extensive open water in the Barents Sea seen in recent winters is expressed as a prominent peak in temperature anomalies, but the ice loss itself seems at least partly related to enhanced inflow of Atlantic derived waters. The point here is that ice loss leads to Arctic amplification, but processes in addition to the direct effects of GHG warming affect the sea ice loss. The argument that the "other" processes can only modulate the sea ice loss induced amplification but cannot cause large AA without sea ice loss is hence not quite right in my opinion.

Response: Thanks for this important comment. We agree that the overall warming in the Arctic is enhanced by the water vapor feedback, poleward energy transport and other processes, besides the direct warming effect of increased CO₂. And this overall warming is largely behind the sea-ice melting. From this view point, these processes can contribute to the sea-ice loss and thus indirectly to the AA. We have added the following text in Discussion on p.11:

"We emphasize that enhanced downward LW radiation³⁴ from increased CO₂, water vapor and other greenhouse gases will lead to Arctic and global warming, which drives the long-term decline of Arctic sea ice. Poleward energy transport is essential for Arctic energy balance, and increased such transport^{16, 34} in a warmer climate would help the Arctic stay in a warmer state, contributing to sea-ice melting. The sea-ice loss in turn causes AA under a warming climate. From this perspective, any external forcing (e.g., increasing CO₂) or internal processes (e.g., water vapor feedback^{12,15}, poleward heat transport¹⁶, a stable lower troposphere¹³) that can cause or enhance Arctic warming may contribute to AA through its impact on sea ice. However, without the LW, SH, LH and other

surface flux changes associated with the sea-ice loss (e.g., after the sea ice melts away or in a model with little sea-ice loss), Arctic warming rate would be similar to the global warming rate, leading to little AA.”

Specific:

Abstract, sentence 1: Be careful with loose terminology. Exactly how are the authors defining Arctic Amplification? Is it the Arctic temperature trend as compared to the global average trend? There are parts of the “rest of the world”, notably around the Antarctic Peninsula, that show a warming trend as large as in the Arctic. And warming over what period, the instrumental record?

Response: Added “(north of 67°N)” to define the Arctic of this study, and inserted “recent” before “observations” to reflect that this is mainly for recent decades. The exact definition of AA is given in the text, it generally refers to the enhanced warming over the Arctic in comparison with the global-mean.

Line 32: For context, it should be noted here that the observations show downward trends in ice extent, strongest at the end of the melt season in September. The way the text is written might give the unintended impression that ice loss is something out there in the future.

Response: Indeed, this study mainly focuses on future sea-ice loss projected by the models under increasing CO₂, although we have added a section on historical (1979-2016) changes (Fig. 1) mainly based on ERA-I data (but also examined NOAA satellite SIC data) in the revised version. The seasonality and spatial patterns of the SIC change during 1979-2016) (Fig. 1) are similar between ERA-I and CMIP5 models.

Line 43: It should be mentioned right here that the study uses simulations from the CMIP-5 archive. Don’t leave the reader wondering.

Response: The various data and simulations used in this study are mentioned immediately following these questions.

Line 45: It might be worth mentioning here that (as I understand it) current emission rates are pretty much in line with RCP 8.5, hence it is not an extreme scenario.

Response: We removed “high” before “emissions”. We have to leave out details like this in a brief article to *Nature Communications* or other *Nature* journals.

Line 46: Be specific with terminology. By saying “ice loss peaks in the latter half of the 21st century” are you saying that the annual rate of change is largest in that period?

Response: Changed to “Arctic sea-ice loss rate peaks around 2070; thereafter ... (Fig. 3b and Fig. S3 in SI)”.

Figure 1: This figure might be much easier to digest if presented as bar plots. For example, for each month, show three bars, of different colors, side by side: the SIC change, Arctic temperature change and AA.

Response: This figure is no longer needed and deleted because the same AA is now shown in the new Fig. 3.

Figure 1a: In the text, briefly discuss why we see the peak in SIC change in July.

Response: We wish we know the reason. This could be due to abnormal seasonality in a few of the 9 models used in this figure. Since this is a minor detail that does not affect our key results, we did not explore it further. When more CMIP5 models used, the seasonality of the SIC loss for the recent period (1979-2016) is highly comparable to ERA-I (and NOAA satellite data) (Fig. 1).

Figure 2: I'm a bit puzzled by this figure. Figure 1 shows AA in terms of the ratio between the Arctic and global average temperature, while Figure 2 (and others that follow) show instead the difference between the global and Arctic temperature. I find this to be confusing. Also, trying to show the temperature difference on the same scale as the energy fluxes is not effective - for example, on the top panel, the energy fluxes range from -30 to +30 W m⁻² while the range for the temperature difference is only from about zero to plus 1.5 so it's really hard to see what is going on. A better presentation is warranted.

Response: We now use Arctic-to-global ratio of Tas change in most of the figures, and (old) Figure 2 is completely redrawn with fewer lines.

Figure 2: For the top panel, the results seem to show that the difference between 2070-2099 and 1970-1999 in the turbulent sensible heat flux peaks at less than 10 W m⁻²; the peak difference for the turbulent latent heat flux is only around 10 W m⁻². These are small values, and would seem to counter the argument that AA is related to enhanced surface heat fluxes associated with sea ice loss. Why are these fluxes so small? Or am I simply missing something?

Response: For Nov-Dec, the SH+LH change from 1970-1999 to 2070-2099 is around 18W/m² (see revised Fig. 3a). Considering that a doubling of atmospheric CO₂ has a forcing of about 4W/m², that number is not small! We are talking about consistent long-term forcing, not year-to-year fluctuations. Following your suggestion, we have added the upward longwave flux change to the figure, and it shows a change of around 55 W/m² for Nov-Dec (Fig. 3a). Thus, the upward LW radiation change is indeed a bigger signal; however, part of that LW increase is associated with the general warming under rising CO₂ (and water vapor). If we take the JJA warming and JJA LW radiation change as the part due to the background warming without AA, then the Nov-Dec minus JJA difference of the LW change would give an extra heating of about 40W/m² that may be attributed to AA, and this number is still much larger than the SH+LH change. In other words, the

extra heating of the air through upward LW radiation is more important than the SH+LH change. We now point out this in the paper (top of p.4).

Figure 2 and elsewhere: As discussed above, I don't think that the net longwave flux is the thing that the authors should be focusing on – the upward longwave flux strikes me as much more relevant as it is a flux of energy into the lowest layer of the atmosphere from the surface.

Response: Upward LW radiation is used in all plots, while SH+LH is combined to one line in all figures. Net LW is removed from the figures.

Figure 3 caption: Should this instead be “40 year periods centered on the plotted year”?

Changed to “Time series of the difference between the 20 year periods separated by the plotted year ...”. The 20 year period is important information.

Figure 4: This figure is entirely undecipherable – way too many overlapping lines. It will be much better to show individual panels for the different variables.

We removed the contours for net energy flux from this Figure. Now it has only the red and yellow contours. It is easier to read. The old version had the black contours for the net energy flux and the black contours were mixed up with black coastal lines, making it hard to read.

Throughout the text: Correlations are cited without corresponding p-values.

The p-values are very significant (<0.01 for all cases) due to large number of data points for both the spatial and temporal correlations. We now mention this in figure captions.

Reviewer #3 (Remarks to the Author):

The paper contrasts Arctic warming and Arctic amplification in periods with substantial sea ice retreat and after the loss of most of the initial sea ice area to determine the role of sea ice and sea ice loss for Arctic amplification. It thus addresses an important problem in Arctic climate research with an interesting and novel idea. Unfortunately, the specific approach taken here may suffer from a fundamental flaw: By contrasting AA in the 21st and 22nd century (with sea ice loss) and the 23rd century (without sea ice loss), the authors compare episodes of global warming that are very different in terms of the radiative forcing applied and the global warming rate. The manuscript does not contain sufficient details on these issues, but they certainly carry the risk to mask the signal the analysis is trying to reveal.

The arguments not affected by these issues, such as temporal and spatial correlations between Arctic warming and sea ice loss, are by themselves not strong enough to support the papers' conclusions. I therefore suggest to repeat the analysis in experiments with abrupt or steady forcing, and make sure that periods with and without sea ice are comparable in terms of the global climate change signal.

Response: To address this concern of changing global warming rate and the concern of Reviewer#1 regarding the potential impact of a changed mean climate state, we have performed two multi-century simulations with and without fixed Arctic sea-ice cover in calculating the surface fluxes under a constant forcing of 1%/yr CO₂ increase using the NCAR CESM1, a fully coupled climate model. As shown in new Fig. 6, the global-mean warming rate is fairly constant over the 235 years of simulation for both cases. These new simulations suggest that the changing mean climate states are not a factor in causing the AA. This is because the mean climate states changed greatly with the CO₂ level varying from the pre-industrial level of 284.7ppm to around 2950ppm at the end of the simulations, yet the dependence of the AA on sea-ice loss remains strong throughout the simulation period. Since the same strong dependence of the AA on sea-ice loss still exists in these simulations with constant CO₂ forcing, the changing forcing used in the CMIP5 simulations is unlikely to be a major factor for the analyzed AA vs. SIC relationship. We hope the new simulations have addressed this concern adequately.

Reviewer #1 (Remarks to the Author):

This revised manuscript asserts that sea ice loss is a key component driving Arctic Amplification and in its absence amplification does not have. Many lines of evidence for this claim are provided and I think that the results justify the claim and the results are convincing. The 1%CO₂ simulations are an excellent addition to this paper and allow the paper to assert causality here. The presented results are novel and are of interest to others in the community.

The authors have done an excellent job addressing my original concerns.

Signed: Patrick Taylor

Reviewer #3 (Remarks to the Author):

This review of the revised paper only covers one issue in the methodology and interpretation of the additional model experiments, which I believe is crucial for the further development of the manuscript and overarches other questions.

Adding dedicated model experiments to show that once the disappearance of sea ice is artificially disabled in the model, Arctic amplification is reduced very substantially would indeed strongly support the key message of the paper, and provide exciting new knowledge about the role of sea ice in Arctic amplification. Unfortunately, the presented experiments fall short of this goal:

By using upward (ice to air) surface fluxes from the control climatology in areas of disappearing sea ice, the authors do not only prescribe the existence of sea ice at these grid points. Seen from the atmosphere, surface temperatures in these areas are effectively fixed, and energy is artificially extracted from the atmosphere. For example, downwelling (atmosphere-ice) longwave radiation increases in a warming experiments. In the real world or a fully coupled model, the surface warms in response, and upward longwave radiation increases as well, and contributes to additional warming of the near-surface atmosphere. This surface coupling is known to play an important role in surface-amplified Arctic amplification of climate change. In the FixedIce setup, the downwelling longwave radiation increases, but the upward radiation remains fixed at its climatological value.

The hypothetical “replacement ice” in the fixed ice run acts as a high-latitude energy sink with infinite heat capacity: No matter how much longwave and shortwave radiation is absorbed at the surface, the energy fluxes from the ice layer to atmosphere and ocean cannot respond.

As the authors find it difficult to directly manipulate the fluxes in the CESM sea ice model, a workaround might be to use a simple formulation of the sea-ice energy balance. A fundamental condition would be that the construction does not fundamentally violate energy conservation and that the surface temperature seen by the atmosphere can respond to warming.

Response to Reviewers' comments for manuscript NCOMMS-17-03941A

Reviewers' comments:

Reviewer #1 (Remarks to the Author):

This revised manuscript asserts that sea ice loss is a key component driving Arctic Amplification and in its absence amplification does not have. Many lines of evidence for this claim are provided and I think that the results justify the claim and the results are convincing. The 1%CO₂ simulations are an excellent addition to this paper and allow the paper to assert causality here. The presented results are novel and are of interest to others in the community.

The authors have done an excellent job addressing my original concerns.

Signed: Patrick Taylor

Response: We thank Dr. Taylor for reviewing the revised manuscript again and for the positive comment on the revised version. We truly appreciate his constructive comments in the first round of review that helped improve the manuscript substantially.

Reviewer #3 (Remarks to the Author):

This review of the revised paper only covers one issue in the methodology and interpretation of the additional model experiments, which I believe is crucial for the further development of the manuscript and overarches other questions.

Adding dedicated model experiments to show that once the disappearance of sea ice is artificially disabled in the model, Arctic amplification is reduced very substantially would indeed strongly support the key message of the paper, and provide exciting new knowledge about the role of sea ice in Arctic amplification. Unfortunately, the presented experiments fall short of this goal: By using upward (ice to air) surface fluxes from the control climatology in areas of disappearing sea ice, the authors do not only prescribe the existence of sea ice at these grid points. Seen from the atmosphere, surface temperatures in these areas are effectively fixed, and energy is artificially extracted from the atmosphere. For example, downwelling (atmosphere-ice) longwave radiation increases in a warming experiments. In the real world or a fully coupled model, the surface warms in response, and upward longwave radiation increases as well, and contributes to additional warming of the near-surface atmosphere. This surface coupling is known to play an important role in surface-amplified Arctic amplification of climate change. In the FixedIce setup, the downwelling longwave radiation increases, but the upward radiation remains fixed at its climatological value. The hypothetical "replacement ice" in the fixed ice run acts as a high-latitude energy sink with infinite heat capacity: No matter how much longwave and shortwave radiation is absorbed at the surface, the energy fluxes from the ice layer to atmosphere and ocean cannot respond.

As the authors find it difficult to directly manipulate the fluxes in the CESM sea ice model, a workaround might be to use a simple formulation of the sea-ice energy balance. A fundamental

condition would be that the construction does not fundamentally violate energy conservation and that the surface temperature seen by the atmosphere can respond to warming.

Response: We thank the reviewer for raising this potentially serious issue in our FixedIce experiment. We have checked and compared the sea-ice concentration (SIC) from the CTL run and from the FixedIce run, and found that the areas over which the CTL fluxes were applied were very small compared with the total Arctic sea-ice cover (see Figure below, which is included as new Fig. S10 in the paper), especially during the early part of the simulation. Please note that we applied the internal sea-ice model-calculated fluxes to the CTL SIC fraction whenever there existed non-zero SIC. Only for the CTL SIC fraction of the gridcells where the initial ice melted away completely, the CTL fluxes were applied (only to the CTL SIC fraction). Because the CTL SIC fraction is low over these gridcells, which account for only a small fraction of the total gridcells with sea ice, the impact of the deficiency of using the CTL fluxes is likely to be small. Over most of the Arctic area covered by sea ice, the internally-calculated fluxes were applied (to the CTL SIC fraction) in the FixedIce run. Thus, the concern raised above does not really apply to our FixedIce experiment. The difference between the standard 1%CO₂ run and the FixedIce run comes primarily from the application of the internal sea-ice model-calculated fluxes to the CTL SIC fraction in the FixedIce run, instead of applying them to the SIC fraction existed at the time inside the sea-ice model as in the standard 1%CO₂ run.

The use of the CTL fluxes over the small fraction of Arctic sea-ice cover is not ideal but necessary because 1) the sea-ice model does not calculate surface fluxes for gridcells without any ice; 2) the internally-calculated surface temperatures for these gridcells are for water surfaces and thus can't be directly used as the temperature for ice surfaces for computing upward LW radiation, SH and LH fluxes; and 3) the calculations of surface fluxes over ice surfaces are complicated. Thus, there is not an easy solution to it. Fortunately, the SIC was low for the gridcells where the CTL fluxes were applied (to the SIC fraction only) and the number of such gridcells was small compared with the total gridcells with ice (Fig. S10). Therefore, we don't think the issue raised in the above comment is a major concern in our FixedIce experiment.

Also, please note that our conclusion regarding the dominant role of sea-ice loss in Arctic amplification was based on many lines of evidence besides the FixedIce experiment, as pointed out by Reviewer #1. Therefore, we are confident that we have provided sufficient evidence to support the conclusion.

We have substantially revised the text on the CESM1 experiments in Methods to reflect most of the points made above. Our initial write-up gave a misleading impression that the CTL fluxes were applied over most of the Arctic, which was not true.

Figure S10. Mean Arctic sea-ice concentration (SIC, %) for September (left) and March (right) from the control run (top row) and the internally-calculated SIC from the FixedIce run averaged for the 20 year period around the 1st (2nd row), 2nd (3rd row) and 3rd (4th row) doubling of atmospheric CO₂. All colored areas have positive SIC where internally-calculated surface fluxes were used in the FixedIce run. Surface fluxes from the control run were used only over the lower latitude sea-ice margins where the initial sea-ice were melted away completely (i.e., only over the CTRL SIC fraction of the areas with color in the top row but in white in the lower rows).

Reviewer #3 (Remarks to the Author):

The authors argue that the replacement of interactive by control fluxes only occurs over a small area and thus should not be a major concern. While I appreciate the addition of Figure S10 that is supposed to support the argument, I am afraid that I am not convinced enough to support publication. Comparing Figures b) and h) shows that in winter, when most of the Arctic amplification occurs and the difference between surface temperatures and fluxes over sea ice and ocean maximises, a substantial ocean area to the east of the control climate ice edge between Svalbard and Scandinavia becomes ice-free, so that the control fluxes would be applied. The same is true for a smaller area around the Bering Strait.

The argument that the non-conservation issue in the FixedIce run is small may well be correct, but in my view, it requires a more rigorous and quantitative support than given by the authors so far. For example, comparing the TOA flux imbalance with ocean heat uptake (using fluxes at the ice-ocean interface or the integrated ocean temperature change per year) in the 1%CO₂ and FixedIce runs could show the global extent of the artefact, and a map of annual mean changes in total surface fluxes between the control and 1%CO₂ runs and between control and FixedIce (where of course the internal fluxes would have to be modified as done in running the model) could show local imbalances. For the latter, some changes in ocean circulation and non-equilibrium heat uptake would be expected in the 1%CO₂ run, but the artificial heat sink introduced in FixedIce could show up at the ice margins and over the newly open ocean. Hopefully, it would be small in extent and magnitude, which should then convince readers (and myself) that this disadvantage of the setup is not a cause for major concern.

Reviewers' comments on Manuscript NCOMMS-17-03941B

We thank the reviewer for his/her further comments (in *italic* below), which motivated us to examine the energy fluxes in more detail, as described below and in further revised paper (mainly in the Suppl. Information).

Reviewer #3 (Remarks to the Author):

The authors argue that the replacement of interactive by control fluxes only occurs over a small area and thus should not be a major concern. While I appreciate the addition of Figure S10 that is supposed to support the argument, I am afraid that I am not convinced enough to support publication. Comparing Figures b) and h) shows that in winter, when most of the Arctic amplification occurs and the difference between surface temperatures and fluxes over sea ice and ocean maximises, a substantial ocean area to the east of the control climate ice edge between Svalbard and Scandinavia becomes ice-free, so that the control fluxes would be applied. The same is true for a smaller area around the Bering Strait.

Response: We agree that by the end of the 235-yr simulation, there are noticeable differences between the control run and the FixedIce run in the areas with and without any sea ice around the original sea-ice margins, as shown by Fig. S10b and Fig.S10h. For example, for the month of March, the control-run fluxes were used over the area south of the Bering Strait in the Pacific during and after years 131-150 (but not during and before years 61-80) as the sea ice there was melted away completely during and after years 131-150 (Fig. S10). Because of this, the internal ice model would not cover this water surface and thus would not calculate the ice-to-ocean and ice-to-atmosphere fluxes over this area, and we had to use the control-run fluxes for this area for the ice-to-ocean and ice-to-atmosphere fluxes when the control-run ice cover was used in the coupler. Please note that the control-run fluxes were used *only* over the areas that are in color for the control run but are in white for the FixedIce run in Fig. S10.

The argument that the non-conservation issue in the FixedIce run is small may well be correct, but in my view, it requires a more rigorous and quantitative support than given by the authors so far. For example, comparing the TOA flux imbalance with ocean heat uptake (using fluxes at the ice-ocean interface or the integrated ocean temperature change per year) in the 1%CO₂ and FixedIce runs could show the global extent of the artefact, and a map of annual mean changes in total surface fluxes between the control and 1%CO₂ runs and between control and FixedIce (where of course the internal fluxes would have to be modified as done in running the model) could show local imbalances. For the latter, some changes in ocean circulation and non-equilibrium heat uptake would be expected in the 1%CO₂ run, but the artificial heat sink introduced in FixedIce could show up at the ice margins and over the newly open ocean.

Hopefully, it would be small in extent and magnitude, which should then convince readers (and myself) that this disadvantage of the setup is not a cause for major concern.

Response: Thanks lot for the helpful suggestions. We have now examined the TOA and surface flux changes and their possible causes in the 1%CO₂ run and the FixedIce run. Here we discuss a few of these additional plots we made, focusing mainly on the Arctic Ocean where the fixed sea-ice cover was imposed in calculating the surface fluxes in the FixedIce run. We focused on the annual-mean changes as suggested by the reviewer, although we did examine the changes for the months of September and March, and the results are similar. We hope the discussion and the new figures, which are added to the Suppl. Information, will convince the reviewer and other readers of the paper that the setup of the FixedIce run did not induce major artifacts for the energy balance in the model.

Fig. R1 below shows the time series of the global-mean and Arctic-mean changes in the TOA and surface net energy fluxes from the two simulations. Over the Arctic (67°-90°N), the TOA fluxes are very similar for the two runs, while the net surface flux into the Arctic Ocean is substantially smaller in the 1%CO₂ run than in the FixedIce run (Fig. R1b). This is expected because the use of the fixed sea-ice concentration (SIC) in the FixedIce run reduces the upward longwave radiation and sensible and latent heat fluxes (see Figs. 7 and 8 in the paper, and Figs. R9-10 below) and therefore should lead to a larger downward net flux in the FixedIce run. Thus, the Arctic surface flux difference shown in Fig. R1b is what we should expect physically from the change we imposed, whose impact on the Arctic TOA flux is negligible. Because of this, the small difference in the global-mean TOA flux between the two runs after year ~100 (Fig. R1a) is likely due to other changes outside the Arctic region, rather than the direct effect of the imposed SIC change in the Arctic. Noticeable differences exist only after about year 100 in the global-mean fluxes between the two runs, and the TOA-minus-surface flux difference is similar between the two runs throughout the whole simulations (Fig. R1a). Thus, the global-mean and Arctic-mean TOA and surface net fluxes from the two runs are reasonable.

To examine the spatial patterns, we compare the change patterns between the two runs for the TOA and surface net energy fluxes in Fig. R2 and Fig. R3, respectively. Again, the TOA flux changes are similar in the two runs, with noticeably higher downward TOA net fluxes over most of the Arctic Ocean in the 1%CO₂ run (Fig. R2e) than in the FixedIce run (Fig. R2f) at the time of the 3rd CO₂ doubling (mainly due to increased absorption of solar radiation in the 1%CO₂ run), except over the areas with large SIC decreases in the FixedIce run by the 3rd CO₂ doubling (see Fig. R2 & Fig. R4). Examination of the changes in the outgoing longwave radiation (OLR) and TOA net shortwave (SW) radiation (not shown) revealed that the increase in the Arctic TOA net flux is mainly due to the larger increase in the absorbed SW radiation than the increase in OLR in the 1%CO₂ run, while it is mainly due to reduced OLR (as surface upward LW radiation is reduced) as the albedo and thus SW changes are small in the FixedIce run. Thus, these TOA flux changes are expected physically based on the imposed the changes.

For the surface net energy flux (Fig. R3), the broad change patterns are similar between the two runs over most of the globe, except the Arctic where the fixed SIC in the FixedIce run is expected to reduce the absorbed SW radiation, upward longwave (LW) radiation, and upward sensible (SH) and latent (LH) heat fluxes in comparison with the 1%CO₂ run (see Figs. 7 and 8 in the paper & Figs. R9-10 below). These reductions in upward fluxes lead to higher downward

net energy fluxes in the FixedIce run than in the 1%CO₂ over the Arctic Ocean (Fig. R3). Further examination revealed that surface downward SW radiation is reduced over the Arctic due to increased cloudiness in both simulations (Figs. R5 and R6); however, due to reduced sea-ice cover in the 1%CO₂ run (Fig. R4), surface absorbed SW radiation is actually increased despite the decreased downward SW radiation in this run (Fig. R7a,b,e). Because the sea-ice cover is fixed in calculating the SW flux in the FixedIce run, the reduced downward SW radiation leads to decreased absorbed SW radiation in the FixedIce run (Fig. R7b,d,f).

As the fixed sea-ice cover reduces upward LW radiation, surface net LW radiative heating increases more in the FixedIce run than in the 1%CO₂ run in the Arctic (Fig. R8). On the other hand, the upward surface LH (Fig. R9) and SH (Fig. R10) heat fluxes increase substantially only in the 1%CO₂ run over the Arctic, and they partially offset the large increases in the absorbed SW radiation (Fig. R7e) and the moderate increase in the net LW radiative heating (Fig. R8e) over the Arctic in the 1%CO₂ run, which results in only moderate changes in the net surface energy flux (Fig. R3e). This differs from the FixedIce run, in which the changes in the LH and SH fluxes are small (Figs. R8f and R9f), and the net surface energy flux change (Fig. R3f) results mainly from the reduced net SW (Fig. R7f) and greatly increased net LW (Fig. R8f) fluxes. These changes in the FixedIce run happened over most of the Arctic, not just over the sea-ice margins (Fig. S10) where the control-run fluxes might be used. Thus, these changes are mainly due to the use of the fixed SIC in calculating the surface fluxes, rather than due to the use of the control-run fluxes over a few areas around the original sea-ice margins. This further demonstrates that the use of the fixed SIC, rather than the use of the control-run fluxes, is the main reason behind these differences between the 1%CO₂ and FixedIce runs. These analyses of the TOA and surface fluxes also suggest that there are no major artifacts resulting from the imposed changes in the FixedIce run.

Please note that we have provided several lines of evidence, besides using the 1%CO₂ minus FixedIce difference, to support our conclusion that sea-ice loss is necessary for large Arctic amplification (AA) to occur. These include the diminishing AA in the 23rd century in the CMIP5 models (Fig. 3 in the paper) and in the CESM1 1%CO₂ run (Fig. 7d) when Arctic sea-ice melting becomes small. Another evidence is the strong correlation between Arctic sea-ice loss and the AA among the 38 CMIP5 models during the 21st century (Fig. 4), and the spatial and seasonal relationship between Arctic warming and sea-ice loss (and the associated surface energy flux changes). Thus, while the results from the FixedIce experiment provide a strong confirmation of the essential role of sea-ice loss, they represent only one of the several lines of evidence presented in the paper.

Fig. R1. Changes (relative to control climatology) in annual (a) global-mean and (b) Arctic-mean net energy fluxes (positive downward) at the top-of-atmosphere (TOA) and surface from the standard 1%CO₂ and fixed_Ice runs.

Fig. R2. Changes (relative to control climatology) in top-of-atmosphere (TOA) annual net energy flux (W/m^2 , positive downward) around the 1st (top row), 2nd (middle row) and 3rd (bottom row) doubling of atmospheric CO₂ from the standard 1%CO₂ (left) and Fixed_Ice (right) runs.

Fig. R3. Changes (relative to control climatology) in annual surface net energy flux (W/m^2 , positive downward) around the 1st (top row), 2nd (middle row) and 3rd (bottom row) doubling of atmospheric CO₂ from the standard 1%CO₂ (left) and Fixed_Ice (right) runs.

Fig. R4. Changes (relative to control climatology) in annual sea-ice concentration (SIC, in % of grid area) around the 1st (top row), 2nd (middle row) and 3rd (bottom row) doubling of atmospheric CO₂ from the standard 1%CO₂ (left) and Fixed_Ice (right) runs.

Fig. R5. Changes (relative to control climatology) in annual surface downward shortwave radiative flux (W/m^2 , positive downward) around the 1st (top row), 2nd (middle row) and 3rd (bottom row) doubling of atmospheric CO₂ from the standard 1%CO₂ (left) and Fixed_Ice (right) runs.

Fig. R6. Changes (relative to control climatology) in annual total cloud cover (%) around the 1st (top row), 2nd (middle row) and 3rd (bottom row) doubling of atmospheric CO₂ from the standard 1%CO₂ (left) and Fixed_Ice (right) runs. The increase in Arctic clouds is seen in low, middle and high clouds.

Fig. R7. Changes (relative to control climatology) in annual surface net shortwave radiative flux (W/m^2 , positive downward) around the 1st (top row), 2nd (middle row) and 3rd (bottom row) doubling of atmospheric CO₂ from the standard 1%CO₂ (left) and Fixed_Ice (right) runs.

Fig. R8. Changes (relative to control climatology) in annual surface net longwave (LW) radiative flux (W/m^2 , positive downward) around the 1st (top row), 2nd (middle row) and 3rd (bottom row) doubling of atmospheric CO₂ from the standard 1%CO₂ (left) and Fixed_Ice (right) runs.

Fig. R9. Changes (relative to control climatology) in annual surface latent heat flux (W/m^2 , positive upward) around the 1st (top row), 2nd (middle row) and 3rd (bottom row) doubling of atmospheric CO_2 from the standard 1% CO_2 (left) and Fixed_Ice (right) runs.

Fig. R10. Changes (relative to control climatology) in annual surface sensible heat flux (W/m^2 , positive upward) around the 1st (top row), 2nd (middle row) and 3rd (bottom row) doubling of atmospheric CO₂ from the standard 1%CO₂ (left) and Fixed_Ice (right) runs.